# Single cell and spatial transcriptomic analyses reveal microglia-plasma cell crosstalk in the brain during *Trypanosoma brucei* infection

Juan F. Quintana [1,2] ✉, Praveena Chandrasegaran[1,2], Matthew C. Sinton [1,2], Emma M. Briggs [1,3], Thomas D. Otto [1,4], Rhiannon Heslop [1,2], Calum Bentley-Abbot [1,2], Colin Loney [5,4], Luis de Lecea [6], Neil A. Mabbott [7] & Annette MacLeod[1,2]

Human African trypanosomiasis, or sleeping sickness, is caused by the protozoan parasite *Trypanosoma brucei* and induces profound reactivity of glial cells and neuroinflammation when the parasites colonise the central nervous system. However, the transcriptional and functional responses of the brain to chronic *T. brucei* infection remain poorly understood. By integrating single cell and spatial transcriptomics of the mouse brain, we identify that glial responses triggered by infection are readily detected in the proximity to the circumventricular organs, including the lateral and 3rd ventricle. This coincides with the spatial localisation of both slender and stumpy forms of *T. brucei*. Furthermore, in silico predictions and functional validations led us to identify a previously unknown crosstalk between homeostatic microglia and *Cd138+* plasma cells mediated by IL-10 and B cell activating factor (BAFF) signalling. This study provides important insights and resources to improve understanding of the molecular and cellular responses in the brain during infection with African trypanosomes.

Chronic infection with *Trypanosoma brucei*, the causative agent of Human African trypanosomiasis (HAT) or sleeping sickness, is associated with extensive and debilitating neuroinflammation[1-4]. Widespread glial cell activation in the CNS, measured by ionized calcium-binding adapter molecule 1 (IBA1) and glial fibrillary acidic protein (GFAP) reactivity, has also been reported in both human brain biopsies and in murine models of infection[2,3,5,6]. There is also extensive infiltration of adaptive immune cells that are thought to be critical mediators of the neuroinflammation induced when the parasites colonise

the CNS[5,7]. However, an in-depth characterisation of the transcriptional responses to infection, in particular that of innate immune cells in the CNS, is lacking.

The application of single-cell RNA sequencing (scRNAseq) has been transformative to understanding brain pathologies such as Alzheimer's disease and has also been recently applied to understand immunological responses to viral infections[8-11]. Nevertheless, a major limitation of scRNAseq is that it cannot preserve the spatial distribution in the tissue of origin. The integration of scRNAseq with spatial

[1]Wellcome Centre for Integrative Parasitology (WCIP), University of Glasgow, Glasgow, UK. [2]School of Biodiversity, One Health, and Veterinary Medicine (SBOHVM), MVLS, University of Glasgow, Glasgow, UK. [3]Institute for Immunology and Infection Research, School of Biological Sciences, University of Edinburgh, Edinburgh, UK. [4]School of Infection and Immunity, MVLS, University of Glasgow, Glasgow, UK. [5]MRC Centre for Virus Research, University of Glasgow, Glasgow, UK. [6]Stanford University School of Medicine, Stanford, CA, USA. [7]The Roslin Institute and Royal (Dick) School of Veterinary Studies, University of Edinburgh, Edinburgh, UK. ✉e-mail: juan.quintana@glasgow.ac.uk

transcriptomics enable us to characterise cellular and tissue responses to infections on regional and global scales. This has been successfully applied to characterise local immune responses to *Mycobacterium tuberculosis*[12] and *M. leprae*[13] and in the heart during viral myocarditis[14]. However, to our knowledge, similar approaches have not been implemented to study tissue responses to protozoan parasites. Here, we present a spatially resolved single-cell atlas of the murine CNS in response to *T. brucei*. This integrative approach led us to identify that glia responses triggered by infection are not limited to the hypothalamus but can be readily detected in close proximity to the circumventricular organs (CVOs), coinciding with the localisation of slender and stumpy forms of *T. brucei*. Furthermore, we identified a previously unknown interaction between homeostatic microglia and *Cd138*+ plasma cells mediated by IL-10 and B cell activating factor (BAFF) signalling. Our spatiotemporal atlas offers novel insights into the interaction between the innate and adaptive immunity during chronic CNS infections and represents a resource to improve our understanding of the molecular and cellular responses triggered in the brain upon infection.

## Results

### Single cell transcriptomic analysis of the mouse hypothalamus over the course of *T. brucei* infection

To resolve the complexity of the different cell types and transcriptional pathways involved in the CNS response to *T. brucei* infection with as much singularity and spatial resolution as possible, we employed a combined single cell (scRNAseq) and spatial transcriptomic approach (Fig. 1A), from samples harvested during the onset of the CNS stage (25dpi) and appearance of neurological symptoms (45dpi) (Fig. 1B and C). The overall inflammation in the brain neuroparenchyma and the meningeal space was confirmed at these time points by histological examination (Supplementary Data 1). To further refine our scRNAseq dataset, we focused on the hypothalamus, given its critical role in controlling circadian behaviour[15,16]. We obtained a total of 13,195 cells with an average of 500 genes/cell and 1,500 transcripts/cell (Materials and Methods, Supplementary Fig. 1). Overall, we identified 11 clusters spanning 8 cell types, including microglia (clusters 0, 1, 6, and 9), oligodendrocytes/B cells (cluster 7), astrocytes (clusters 2 and 5), T cells (cluster 3), and vascular-associated cells including endothelium (cluster 4), pericytes (cluster 8), and ependymocytes (cluster 10) (Fig. 1D and Supplementary Data 2 and 3). The microglia subclusters were dominated by the expression of putative markers including *C1qa, Lyz2, Aif1,* and *Cx3cr1*[17], whereas the astrocyte cluster was characterised by the expression of bona fide markers of mature astrocytes, including *Gfap* and *Agt* (Fig. 1E, Supplementary Fig. 2, and Supplementary Data 2)[18]. The vascular-associated cells were further divided into three *Cldn5*+ endothelial cell subclusters, two clusters representing *Pdgfrb*+ pericytes/tanycytes, one cluster representing *Acta2*+ pericytes, and one *Ccdc153*+ ependymocyte cluster (Fig. 1E, Supplementary Fig. 2, and Supplementary Data 4). These data are in agreement with the diversity of the glial cell types previously reported in healthy mouse hypothalamus[19,20].

Disease state analysis revealed differential distribution of cells within the microglia, B cell, and T cell clusters in infected samples compared to naïve controls (Fig. 1F). Furthermore, by computing in silico gene module score to assess the global expression level of inflammatory mediators (e.g., cytokines and chemokines), we identified that responses to chronic *T. brucei* infection (at both 25 and 45dpi) were largely observed in the microglia subclusters (in particular microglia 1 and 2) and, to a lesser extent in endothelial cells, T cells, and adaptive immune cells and were significantly higher than naïve controls (ANOVA, $p < 2.2^{-16}$) (Fig. 1G). Taken together, our data demonstrate that *T. brucei* infection in the CNS induces an inflammatory response predominantly in microglia, as well as T and B cells, and to a lesser extent in endothelial cells. Notably, we did not detect *T. brucei*

enough reads in our scRNAseq dataset for downstream analysis, perhaps owing to the low parasite burden in the brain parenchyma; however, we were able to detect them in our spatial transcriptomics dataset (see below).

### Spatial transcriptomics reveals both *T. brucei* long slender and stumpy forms predominantly in the circumventricular organs

When analysing our spatial transcriptomics datasets, we noted the expression of multiple *T. brucei*-specific genes in infected samples, especially at 45dpi (Fig. 2A and Supplementary Data 5). The majority of the *T. brucei*-specific genes were distributed in discrete locations in the mouse forebrain. For example, *Tb927.6.4280* (*GAPDH*), typically associated with slender forms[21–23], was highly expressed in spatial clusters 0, 1, 4, 10, 14, and 17, that define the anatomical regions corresponding to cerebral caudoputamen or corpus striatum, thalamus, hippocampus, cerebral cortex, hypothalamus, and the circumventricular organs (CVOs; including the lateral ventricle and the 3rd ventricle) (Fig. 2A, B, and Supplementary Data 5). Similarly, Tb927.7.5940 (*PAD2*) was restricted to cluster 17 (Fig. 2A, B, and Supplementary Data 5). The localisation of parasites in these brain regions at 45dpi coincided, at least partly, with an increase in the expression of inflammatory mediators in several brain regions, including around cluster 4 and 17 (both CVOs-related clusters) (Fig. 2C). The spatial distribution of the different developmental stages of *T. brucei* was further confirmed using smFISH against parasite-specific marker genes associated with slender (*GAPDH* and *PYK1*) or stumpy (*PAD2* and *EP1*) life cycle stages[21,24] (Fig. 2D) and by independent histological scoring (Supplementary Data 5). These observations confirm that in addition to passage through the blood-brain barrier, African trypanosomes also exploit the CVOs as points of entry into the CNS[25,26].

To provide insights into the potential diversity of brain-dwelling trypanosomes, including the presence of various developmental stages, we performed gene ontology and pathway analysis on the most abundant *T. brucei* transcripts based on their relative spatial distribution (Fig. 2E and Supplementary Data 5). Overall, we observed an overrepresentation of genes typically associated with metabolically active parasites in the CVOs, such as protein translation ($p\ adj = 3.35^{-57}$) and biosynthetic processes ($p\ adj = 1.05^{-24}$), irrespective of their spatial distribution (Fig. 2E and Supplementary Data 6)[21–23]. Additionally, the transcriptome of the parasites in the CVOs was dominated by genes pathways broadly associated with translation ($p\ adj = 9.14^{-22}$), control of gene expression ($p\ adj = 2.23^{-04}$), and biosynthetic processes ($p\ adj = 3.46^{-08}$), indicating that the CVO-dwelling parasites are metabolically active. Notably, the parasites in the CVOs also expressed genes considered critical regulators of parasite differentiation, such as RBP7A and RBP7B (encoded by *Tb927.10.12090* and *Tb927.10.12100*, respectively), PAD1 (encoded by *Tb927.7.5930*), and PAD2 (encoded by *Tb927.7.5940*) (Supplementary Data 6)[24,27,28]. Together, these results provide an overview of the spatial distribution of African trypanosomes in the mouse forebrain and support the hypothesis that most brain-dwelling trypanosomes display features of replicative slender forms, protein translation, and control of gene expression, together with differentiation commitments in the CVOs.

### Infection-associated mononuclear phagocytes and border-associated macrophages occupy spatial niches in proximity to the ventricular spaces

Having established that microglia display a high inflammatory score upon infection, and *T. brucei* slender and stumpy forms are found in or surrounding CVOs, we next asked whether the spatial distribution of different microglia cell clusters correlates to the distribution of parasites in the forebrain. After subclustering, we identified six discrete myeloid subclusters that displayed a marked differential gene expression signature (Fig. 3A) encompassing a total of 6305 cells across replicates and conditions. For example, cluster 0 and cluster 1

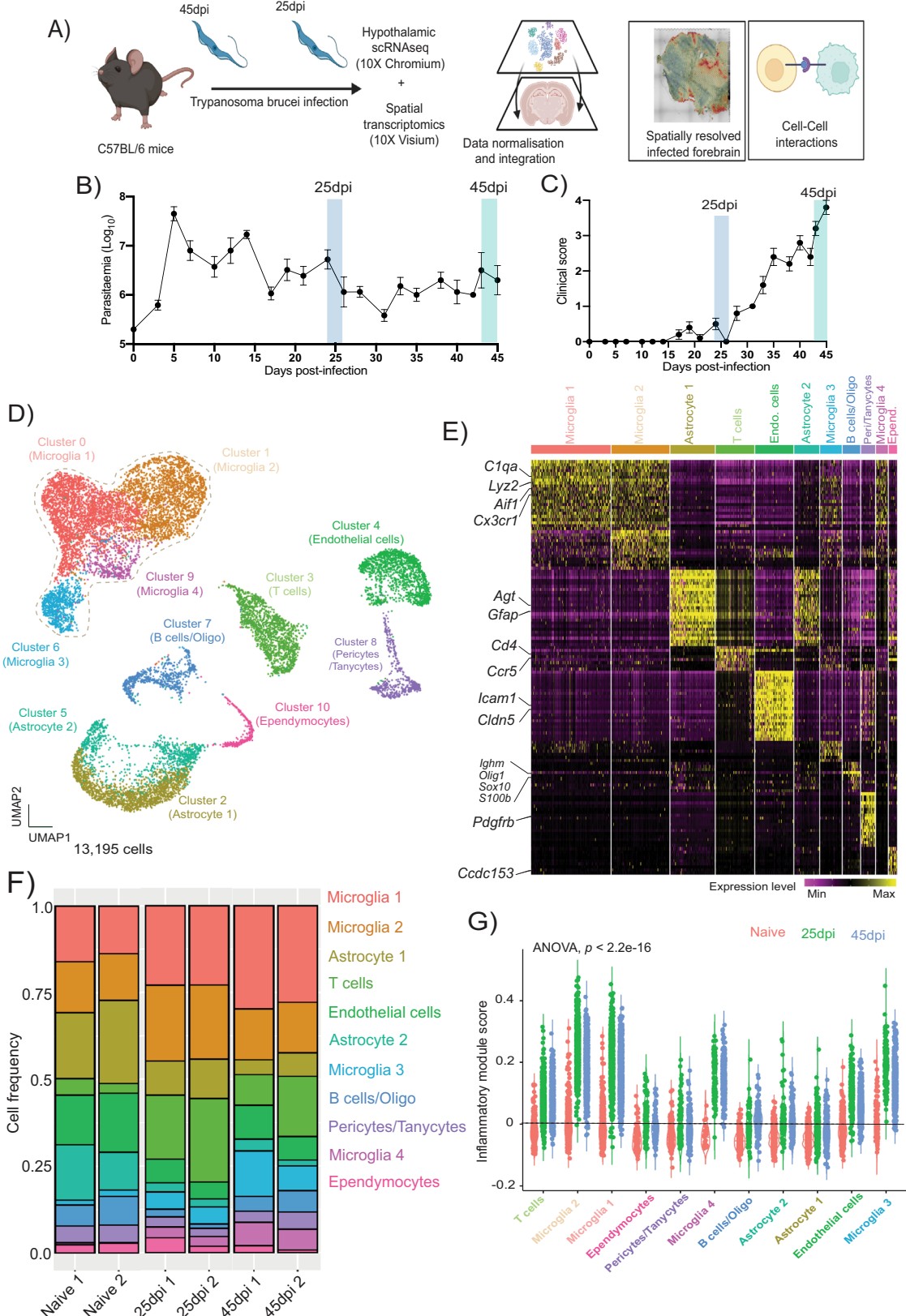

express high levels of putative homeostatic microglia marker genes including *P2ry12, Trem119*, and *Cx3cr1* and may correspond to different homeostatic subsets. Cells in cluster 0 express high levels of *Tgfbr1, Ifngr1*, and *Il6ra*, whereas cluster 1 expresses high levels of *Il10ra* (Fig. 3B, Supplementary Data 7 and 8), suggesting a potential divergent response to cytokine signalling (e.g., interferon gamma (IFNγ) in

cluster one versus IL-10 in cluster 2). Cluster 2 is characterised by the expression of monocyte-specific lineage markers, including *Cd14, Ccrl2, Fcgr2b*, and several MHC-II associated molecules (*H2-Aa1, H2-Ab1*) (Fig. 3B, Supplementary Data 7 and 8). Cluster 3 expresses putative marker genes associated with border-associated macrophages such as *Lyz2, Ms4a7, Ms4a6c, Tgfbi, H2-Ab1, and Lyz2*[29,30], as well as gene

**Fig. 1 | Diversity of hypothalamic glial cells during chronic *T. brucei* infection.**
**A** Overview of the experimental approach applied in this work (created with BioRender.com). Upon infection, the levels of parasitaemia (in $\log_{10}$ scale **B**) and clinical scoring (**C**) were measured in infected animals ($n = 5$ animals per experimental group). The data in **B** and **C** represent the average and standard deviation for both parasitaemia and clinical scores from two independent experiments.
**D** Combined UMAP plot for 13,195 high-quality cells coloured by cell type across all the biological replicates. **E** Heatmap representing the expression level of the top 25 marker genes for each of the cell clusters identified in the combined dataset.
**F** Frequency of the different cell types detected in the murine hypothalamus at the three experimental groups analysed in this study. **G** Inflammatory gene module score of genes typically associated with inflammation across all the cell types detected in (D), split and colour-coded by time point of infection. Statistical analysis using analysis of variance (ANOVA) for multiple comparison testing ($p = 2.2^{-16}$) is also reported.

sets characteristic of anti-inflammatory responses, such as *Mrc1* (encoding for CD206), *Chil3, Arg1, Il1rn, Il18bp*, and *Vegfa* (Fig. 3B, Supplementary Data 7 and 8), indicative of an anti-inflammatory phenotype. Lastly, cluster 4 and 5 express high levels of *Aif1* as well as canonical pro-inflammatory chemokines and mediators of innate immunity (*Ccl5, Mif, Cxcl13*), components of the complement cascade (*C1qa, C1qb*), antigen processing and presentation genes (*H2-Ab1, H2-Eb1*), several interferon-stimulated genes (*Ifitm3, Ifih1*), low or undetectable levels of putative homeostatic microglia markers such as *Tmem119, P2ry12, Sall*, and *Cx3cr1*, and a marked upregulation of genes previously reported in disease-associated microglia (*Apoe, Itgax, Trem2, Cst7*)[31–33] (Fig. 3B, Supplementary Data 7 and 8). [50,51]Based on these results we catalogued clusters 0 to 5 as follow: Homeostatic microglia (HM) 1 (1,688 cells; 26.77%), HM 2, (1,548 cells; 24.55%), *Cd14*⁺ Monocytes (1,396 cells; 22.14%), *Mrc1*⁺ border-associated macrophages (*Mrc1*⁺ BAMs – 812 cells; 12.6%), Infection-associated mononuclear phagocytes (IAMNP) 1 (587 cells; 9.31%), and IAMNP 2 (274 cells; 4.34%) (Fig. 3C and D). Notably, HM 1 and 2, and *Cd14*⁺ monocytes accounted for ~73% of all the microglia detected under homeostatic conditions, but IAMNP 1 and 2, and *Mrc1*⁺ BAMs subclusters progressively increased in frequency over the course of infection, suggesting an adoption of an infection-associated phenotype (Fig. 3C, D). In the spatial context, we found that the gene expression of *Aif1, Adgre1*, specific marker genes for the IAMNP⁺ 1 and IAMNP⁺ 2 subclusters, were highly expressed around the hippocampus, CVOs, and caudoputamen at 25dpi and 45dpi compared to naïve controls (Fig. 3E). Similarly, *Arg1* and *Chil3*, putative marker genes for *Mrc1*⁺ BAMs, were predominantly located in the lateral ventricle and the dorsal 3ʳᵈ ventricle in the infected brain (Fig. 3E), further corroborated by immunofluorescence analysis on independent brain sections (Fig. 3F). Together, our combined analyses demonstrate that infection-associated myeloid subsets (IAMNP 1, IAMNP 2, and *Mrc1*⁺ BAMs) are detected in regions proximal to the CVOs, coinciding with the spatial distribution of trypanosomes and suggesting a functional compartmentalisation of the myeloid subsets in responses to infection.

### Myeloid responses to *T. brucei* infection share common transcriptional features with neurodegeneration diseases

To gain a more comprehensive understanding of microglia responses to infection at the molecular level, we analysed the differentially expressed genes (DEGs) of microglia subtypes in response to *T. brucei* infection, defined as genes with a Log₂ fold change >0.25 or < −0.25 and an adjusted *p* value <0.05. Most of the upregulated DEGs were detected in the HM 1, HM 2, and *Cd14*⁺ subclusters (Fig. 3G). These were associated with an upregulation of MHC-II-mediated antigen presentation (i*H2-Aa1, H2-Ab1*), monocyte chemotaxis (*Ccl2, Ccl4*), adaptive immune responses (*Cd274, Mif, Tnfsf13b*), and responses to IFNγ (*Ifitm3, Aif1*) (Fig. 3H top and Supplementary Data 9). As the infection progresses, we noted an enrichment for genes associated with neurodegenerative disorders (e.g., Amyotrophic lateral sclerosis, Huntington disease, Parkinson disease, and Alzheimer's disease) such as *Apoe, Trem2, and Psen2* (Fig. 3H **left**, Supplementary Datas 9 and 10). These cells also downregulate homeostatic processes associated with organ development (*Tgfbr1, Mertk, Fos*), neurone homeostasis (*Cx3cr1, Itgam*), and responses to cAMP (*Fosb, Junb*) (Fig. 3H, Supplementary Datas 9 and 10). We did not include cells within the IAMNP subclusters

as the overall cell proportion is reduced in naïve compared to infected mice, potentially confounding DEG analysis. Instead, we analysed these two subclusters separately to better understand their transcriptional features. We found that the cells within the IAMNP1 subcluster display the expression of gene pathways broadly associated with antigen processing and presentation (*H2-Ab1, H2-Aa, H2-Eb1*), neutrophil activation (*Ccl5, Fcgr4, Fcer1g*), and synaptic pruning (*C1qc, C1qb, C1qa*), whereas those within IAMNP 2 also upregulate genes associated with translational activity (*Rps5, Rps14, Rpsa, Rps15*), suggesting a transcriptionally active state (Fig. 3I and Supplementary Data 11).

Overall, our data demonstrates a dynamic response of the microglia during *T. brucei* infection; during the onset of the CNS stage (25dpi), homeostatic microglia upregulate transcriptional programmes associated with antigen processing and presentation and development of adaptive immune responses, whilst downregulating genes associated with homeostasis. As the infection progresses (45dpi), the microglia signatures share many commonalities to those identified in neurodegenerative disorders (e.g., *Apoe, Trem2*)[31–34], coinciding with the development of clinical symptoms in these animals. These data suggest a common transcriptional response to inflammatory processes in the CNS. Other myeloid cell types such as *Cd14*⁺ monocytes and *Mrc1*⁺ BAMs constitute additional responders to the infection, albeit with opposing effects; *Cd14*⁺ monocytes and *Mrc1*⁺ BAMs display pro- and anti-inflammatory phenotypes, respectively.

### Chronic *T. brucei* infection recruits follicular-like regulatory *Cd4*⁺ T cells and cytotoxic *Cd8*⁺ T cells into the CNS

We next sought to characterise the population of adaptive immune cells identified in our dataset (1,436 cells in total across samples). We identified three T cell subclusters based on the expression of putative T cell marker genes such as *Trac* and *Cd3g* (Fig. 4A to C, and Supplementary Data 12). Cluster 0 (633 cells; 44.1%) was discarded owing to the lack of identifiable marker genes. The remaining 803 cells (55.9%) were classified correspond to three broad clusters. Cluster 1 (272 cells; 18.9%) and 2 (266 cells; 18.5%) express marker genes associated with cytotoxic T cells such as *Cd8a* and *Gzmb* (Fig. 4A to C, and Supplementary Data 12). Cells within cluster 2 also express high levels of genes associated with cytotoxic T cell activation and effector function (*Ccl5, Klrd1*), a gamma TCR receptor subunit (*Trgv2*), interferon-stimulated genes (*Ifitm1*), and high levels of *Cd52* which is involved in T cell effector function (Supplementary Data 12)[35]. This suggests that cells in cluster 2 potentially represent a specialised cytotoxic T cell subset. Lastly, in addition to *Cd4*, cells in cluster 3 (265 cells; 18.5%) express high levels of genes associated with regulatory CD4⁺ T cells including surface markers (*Cd5, Ctla4, Icos, Cd274*), transcription factors (*Mxd1, Izkf2*), and effector molecules (*Il10, Areg, Il21*) (Fig. 4A to C, and Supplementary Data 12). Notably, these regulatory *Cd4*⁺ T cells also express high levels of marker genes typically associated with follicular helper T cells such as *Maf* and *Slamf5* (Fig. 4A to C, and Supplementary Data 12)[36–38]. These follicular-like regulatory CD4⁺ T cell subsets have been postulated as critical regulators of adaptive responses in lymphoid organs[39–41], but so far have not been reported in the brain during infections. These populations seemed dynamic over the course of infection, with chronic stages associated with a 1.27- and 1.61-fold increase in the abundance of *Cd4*⁺ T cells compared to other subclusters (23.64%, 30.21%, and 38.1% in naïve, 25dpi, and 45dpi,

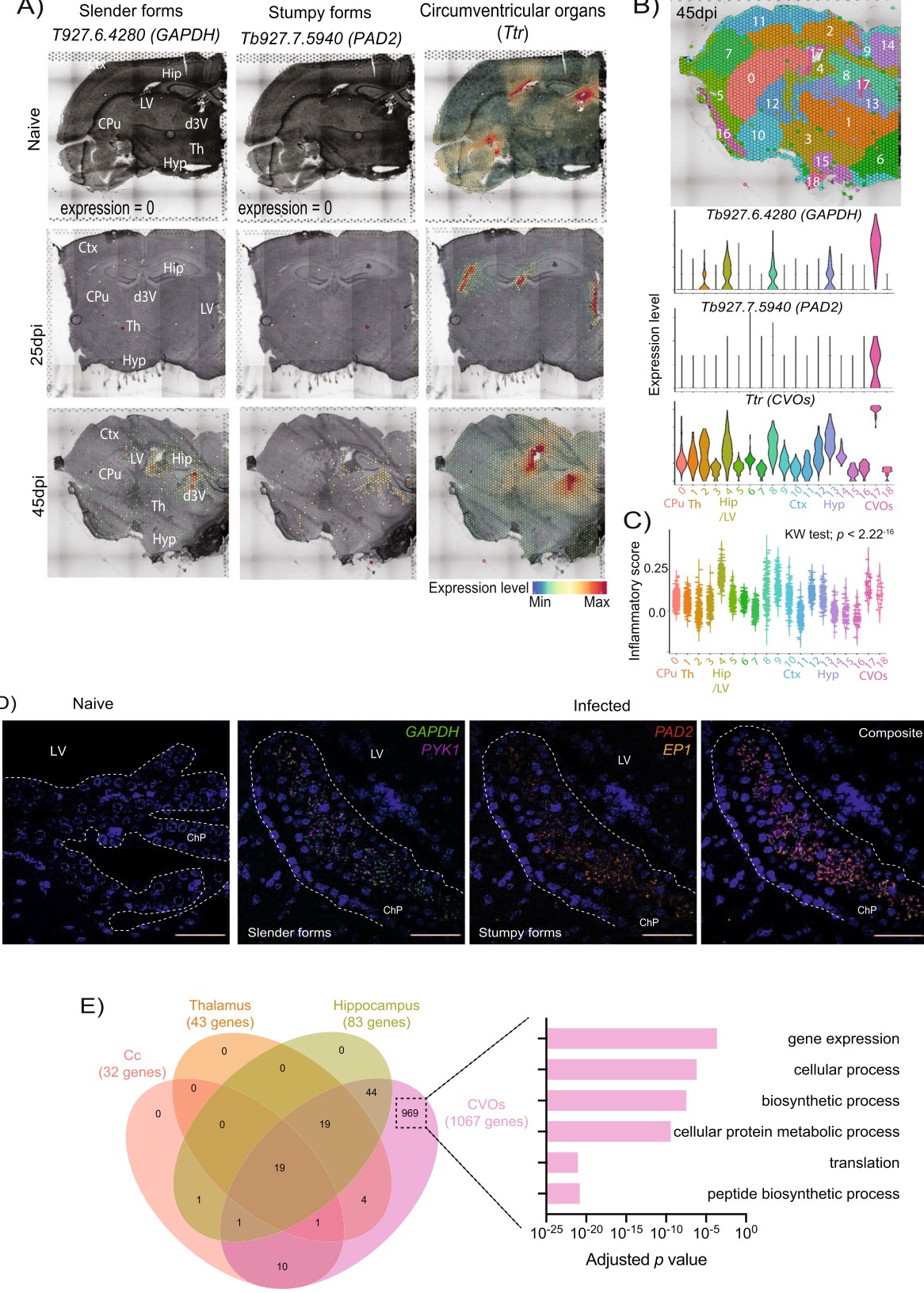

respectively) (Fig. 4B and D), consistent with previous reports[5,42]. Of note, the subcluster identified as cluster 2 *Cd8*+ T cells (*Cd8*+ 2T cells) was only detected in infected samples but not in naïve controls (35.98% and 36.5% at 25 and 45dpi, respectively) (Fig. 4B), indicating a disease-associated T cell subset in the brain. When compared across the brain in the spatial transcriptomics data set, the *Cd4*+ T cell subcluster was mostly detected in the lateral ventricle, external capsule, and the caudoputamen in both naïve and infected samples, whereas the *Cd8*+ T cell subsets showed a more widespread distribution in the brain parenchyma (Fig. 4E).

Cell-cell interaction analysis using NicheNet[43] revealed a network of molecular communication between T cells, stromal cells, and the

**Fig. 2 | Spatially resolved transcriptomics of brain-dwelling *Trypanosoma brucei*. A** In silico projection of the *T. brucei*-specific genes Tb927.6.4280 (*GAPDH*) and Tb927.7.5940 (*PAD2*) and in the spatial transcriptomics from naïve (top row), 25dpi (middle), and 45dpi (bottom) coronal brain datasets. The circumventricular organs (CVOs)-specific marker Transthyretin (*Ttr*) is included as reference. **B Top:** spatial feature plot depicting 18 different transcriptional clusters at in the murine forebrain from infected (45dpi) samples. **Bottom:** Violin plot depicting the relative expression of Tb927.6.4280 (*Gapdh*; slender) and Tb927.7.5940 (*Pad2*; stumpy) in the transcriptional clusters in the infected murine brain. *Ttr* is also included to depict that cluster 17 corresponds to CVOs. Additional regions are also indicated, including caudoputamen or corpus striatum (clusters 0 and 8), hippocampus (cluster 4), Amygdala (cluster 10), and CVOs (cluster 17). The full list of *T. brucei* genes detected can be found in **S3B Table**. **C** Inflammatory gene module score of genes typically associated with inflammation across the different spatially resolved transcriptional clusters at 45dpi. Kruskal-Wallis test for multiple comparisons was

applied and identified significant differences using the normalised gene expression (basemean) as reference. Asterisk denotes significant differences of $p < 0.05$. **D** Representative smFISH probing putative slender markers (*GAPDH* and *PYK1*; middle panel) and stumpy marker genes (*PAD2* and *EP1*; right panels), around the lateral ventricle (LV) and choroid plexus (ChP) in naïve animals (left panels) and infected brain samples (right panels). A composite merging both stumpy and slender markers is also included. Scale bar = 50 μm. The results presented here are representative from three independent experiments. **E** Ven diagram of the different *T. brucei*-specific transcripts detected in several brain regions at 45dpi, based on the spatial distribution shown in 4B. Top 10 GO terms that characterise brain-dwelling African trypanosomes located in the CVOs. The GO terms were chosen using significant genes (defined as $-0.25 < Log_2$ fold change $>0.25$; adjusted $p$ value $<0.05$ using the non-parametric Wilcoxon rank sum test). Ctx cerebral cortex, CPu caudoputamen, Hip Hippocampus, Am Amygdala, CVOs circumventricular organs, including the lateral ventricle (LV) and the dorsal 3rd ventricle (d3V), Th thalamus.

vasculature, in infected samples compared to naïve controls (Fig. 4F, G). For instance, ependymocytes and endothelia cells express high levels of *Cxcl10* and *Cxcl12*, respectively, which are critical mediators of lymphocytic recruitment into the brain parenchyma during neuroinflammation[44–46]. Microglia also express additional subsets of chemokines (*Ccl2, Ccl3, Ccl4*) with no overlap to those detected in vascular-associated cells (Fig. 4F, G), potentially indicating non-redundant mechanisms of T cell recruitment into the brain parenchyma during infection. Furthermore, the endothelial cells and microglia expressed high levels of cell adhesion markers including *Icam1, Icam2*, and *Pecam1* (Fig. 4F, G), which mediate immune cell transendothelial and extravascular tissue migration[42,47,48]. We also detected additional mediators of T cell activation, including endothelial cell-derived *Il15* and astrocyte-derived *Il18*, which are involved in T cell activation and enhancement of IFNγ production[49–53] (Fig. 4F, G). Together, our data provide an overview of the T cell diversity in the CNS during chronic *T. brucei* infection, including regulatory *Cd4*+ T cells that accumulate in the brain over the course of *T. brucei* infection. Moreover, ligand-receptor mediated cell-cell communication suggests that microglia, ependymocytes, endothelial cells, and astrocytes are involved in the recruitment and activation of T cells into the brain during chronic *T. brucei*, albeit through divergent signalling molecules.

### *Cd138*+ plasma cells are detected in the mouse brain during chronic *T. brucei* infection

Next, we characterised the cells contributing to the genes expressed in cluster 7 (Fig. 1D and Supplementary Data 13). This appeared to represent a heterogeneous grouping of cells expressing high levels of oligodendrocyte markers (*Olig1, Sox10*, and *S100b*) and bona fide B cell markers (*Cd79a, Cd79b, Ighm*) (Fig. 1D, Supplementary Data 2 and 13). Dimensional reduction analysis after subsetting the 648 cells within cluster 7 led us to identify five clusters identified as follow: clusters 1 (225 cells; 34.7%) and 2 (74 cells; 11.41%) expressed high levels of *Olig1* and *Pdgfra* and corresponded to oligodendrocytes, cluster 3 (55 cells; 8.48%) corresponds to *Epcam*+ neuroepithelium, and cluster 4 (39 cells; 6.02%) composed of *Map2*+ neurons (Fig. 5A and Supplementary Data 13). Lastly, in addition to *Cd79a* and *Cd79b* (which encode for the B cell receptor), cluster 0 (255 cells; 39.3%) was also characterised by high expression levels of putative markers associated with plasma cells, including surface markers (*Sdc1* or *Cd138, Slamf7*) and plasma cell-specific transcription factors (*Prdm1, Xbp1, Irf4*) (Fig. 5A, B, and Supplementary Data 13). These cells also express genes associated with regulatory function, including *Il10* and *Cd274* (Fig. 5A, B, and Supplementary Data 13), and was thus labelled as *Cd138*+ plasma cells. Furthermore, the *Cd138*+ plasma cells were detected at low levels in naïve controls (~8% of the cells in this cluster) but increased over the course of infection at 25dpi (61.6%) and 45dpi (88%) (Fig. 5C, D). The enrichment of *Cd138*+ plasma cells during chronic infections was

further confirmed by flow cytometry on independent in vivo experiments, mirroring the proportions detected by scRNAseq (Fig. 5E, F). Notably, *Cd138*+ plasma occupied discrete niches in the naïve brain around the CVOs (dorsal 3rd ventricle) and subthalamic regions (Fig. 5G), but were preferentially detected in the external capsule, corpus callosum, and lateral ventricle at 25dpi, or in the leptomeninges, cingulate cortex, lateral ventricle, and dorsal 3rd ventricle at 45dpi (Fig. 5G). Taken together with the flow cytometry findings, these data suggest a potential expansion of this population in the CVOs and proximal regions. Furthermore, the predicted expression of *Il10*, an anti-inflammatory cytokine shown to be expressed in B cells with a regulatory phenotype[54–56], was tested and confirmed by ELISA of ex vivo stimulation brain-dwelling B cells from infected mice (Fig. 5H), corroborating the in silico data and indicating a regulatory phenotype. Together, these data show the presence of *Cd138*+ plasma cells with a regulatory phenotype in the CVOs and leptomeninges in the murine brain during chronic *T. brucei* infection.

### *Cd138*+ plasma cell supernatant suppresses microglia polarisation towards an inflammatory phenotype

Our scRNAseq data indicates that microglia in the forebrain of *T. brucei*-infected mice express both *Il10ra* and *Il10rb* (which together form the functional IL-10 receptor[57–59] Fig. 5I), and that brain-dwelling *Cd138*+ plasma cells produce IL-10 when stimulated ex vivo (Fig. 5H). Thus, we hypothesised that plasma cell-derived IL-10 may play a role in modulating pro-inflammatory responses in microglia. In silico spatial ligand-receptor interaction analysis around the CVOs identified several significant ligand-receptor interactions upregulated during infection, including *Clec1b-Klrb1c*, involved in regulating NK cell-mediated cytolytic activity[60], and *Lpl-Lrp2*, which are involved in *ApoE*-mediated cholesterol intake in neurons[61] (Supplementary Fig. 5). Additionally, we also identified a robust co-expression of *Il10* and *Il10ra* in the CVOs and leptomeninges (Fig. 5J, Supplementary Fig. 6A), coinciding with the predicted localisation of *Cd138*+ plasma cells at 45dpi, which was independently validated using single molecule fluorescence in situ hybridisation (smFISH) (Fig. 5K). This confirmed that expression of *Il10* in brain-dwelling *Cd138*+ plasma cells and *Il10ra* in homeostatic microglia. Next, we hypothesised that the supernatant from stimulated *Cd138*+ plasma cells might also block microglia polarisation towards a pro-inflammatory state. As expected, BV2 microglia-like cells exposed to *E. coli* LPS for 24 h expressed high levels of the pro-inflammatory cytokines *Il1β* and *Tnfα* (Fig. 5L), which was abrogated when BV2 cells were exposed to supernatant from *Cd138*+ plasma cells (Fig. 5L). Moreover, pre-treatment of the *Cd138*+ plasma cells supernatant with a blocking antibody against IL-10 restores the expression of *Il1β* and *Tnfα* in BV2 microglia, strongly indicating that IL-10 is a key plasma cell-derived anti-inflammatory modulator. Taken together, these data suggest a functional interaction between *Cd138*+ plasma cells and microglia-mediated, mediated at least partially, by IL-10 signalling.

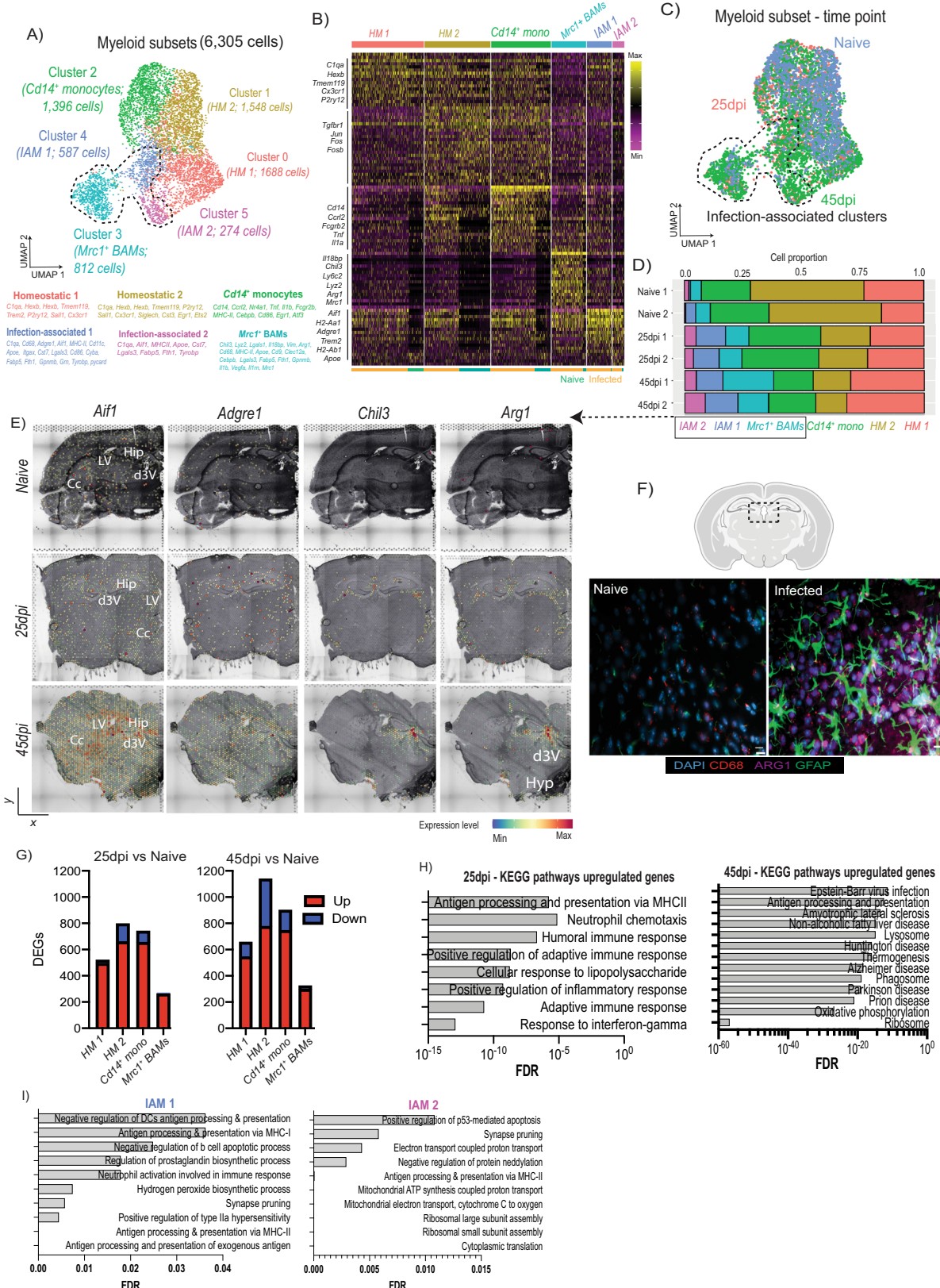

### Homeostatic microglia express the B cell pro-survival factor B cell activation factor (BAFF) signalling

Having established that microglia, T cells and plasma cells are associated with chronic *T. brucei* infection, we next decided to evaluate relevant cell-cell interactions based on the expression level of canonical ligand-receptor pairs. Of these, we observed a network of

complex molecular communication between plasma cells and microglia. A group of ligands were redundantly detected in microglia from HM 1, HM 2, and *Mrc1*⁺ BAMs (*Vcam1*, Spp1, and *Agt*) (Fig. 6A), whereas a second subset of ligands displayed a more cell-restricted expression profile. For example, the pro-survival factor *Tnfsf13b* (encoding for the B cell survival factor, BAFF) was abundantly expressed upregulated by

**Fig. 3 | Diversity and spatial distribution microglia, monocytes, and border-associated macrophages during chronic *T. brucei* infection. A Top panel;** UMAP plot depicting the six subclusters identified as microglia, including the total number of cells in this plot. The dotted line represents the clusters preferentially detected in infected samples compared to naïve controls. **Bottom panel;** selected marker genes for each of the myeloid subsets. **B** Heatmap representing the expression level of putative microglia marker genes for each of the microglia and myeloid subclusters. The cell origin within each cluster (Naïve in teal, infected in orange) is also indicated at the bottom of the heatmap. **C** As in (A) but depicting the identified different time points. The dotted line represents the clusters preferentially detected in infected samples compared to naïve controls. **D** Cell type proportion of the various microglia subclusters detected in Fig. 1E over the course of infection. **E** In silico projection of top marker genes for the infection-associated clusters, including *Aif1, Adgre1, Arg1,* and *Chil3* from naïve (top), 25dpi (middle), and 45dpi (bottom) coronal mouse brain sections. Specific brain regions are also indicated. **F** Imaging analysis of *Mrc1*⁺ BAMs in proximity to the lateral ventricle of naïve and infected mice using immunofluorescence staining for the detection of CD68 (pan-microglia marker) and ARG1 (BAM specific marker). DAPI was included as nuclear staining, and GFAP as a marker for astrocyte reactivity. Scale = 25 μm. The results presented here are representative from two independent experiments. **G** Bar plot indicating the total number of differentially regulated genes (DEGs) at 25dpi (left) and 45dpi (right) compared to naïve controls. Upregulated genes are indicated in red, and downregulated genes are indicated in blue. These genes were defined as having a −0.25 < Log₂ Fold change <0.25, and an adjusted *p*-value of <0.05 using the non-parametric Wilcoxon rank sum test. **H** KEGG gene pathways overrepresented in cluster 2 at 25 and 45dpi. **I** Pathway analysis of the IAMNP subsets based on their individual marker gene profile.

homeostatic microglia upon infection, whereas the expression of its cognate receptor *Tnfrsf17* (or B cell maturation antigen, BCMA) was highly expressed in *Cd138*⁺ plasma cells from infected animals (Fig. 6B and C). Furthermore, the expression of *Tnfsf13b* was higher in microglia from infected mice (Fig. 6D), suggesting that this B cell pro-survival factor is induced upon infection. Spatial ligand-receptor interaction analysis based on co-expression revealed that the co-expression of both the gene for the pro-survival factor, *Tnfsf13b* and the gene for its receptor, *Tnfrsf17*, was restricted to the CVOs and leptomeninges (Fig. 6E and Supplementary Fig. 6B), as identified for the *Il10-Il10ra* ligand-receptor pair. The expression pattern for these two genes was independently confirmed by smFISH analysis and showed that plasma cells expressing *Tnfrsf17* were in close proximity to microglia expressing *Tnfsf13b*, particularly in the vicinity of the lateral ventricle in the brains of *T. brucei*-infected mice (Fig. 6F and Supplementary Fig. 6A and B). The expression of BAFF in microglia upon infection was further analysed by flow cytometry experiments (Fig. 6G), corroborating the in silico predictions. Together, our data suggest that crosstalk between microglia and *Cd138*⁺ plasma cells; In this context, homeostatic microglia promote *Cd138*⁺ plasma cell survival via BAFF, and in turn *Cd138*⁺ plasma produce IL-10 to dampen down inflammatory responses in microglia during *T. brucei* infection (Supplementary Fig. 6C).

## Discussion

To address fundamental questions regarding the innate and adaptive immune responses of the CNS to unresolved, chronic *T. brucei* infection, this study had three main goals: (i) to characterise the temporal transcriptional responses of glial and recruited immune cells to the CNS using single-cell transcriptomic, (ii) to understand the spatial distribution of candidate cell types from the scRNAseq dataset using 10X Visium spatial transcriptomics, and (iii) to model cell-cell interactions taking place in the CNS during chronic infections based on putative ligand-receptor interactions at both single cell and spatial level. Our combined atlas provides novel and important insights for future analyses of the innate and adaptive immune response to chronic CNS infection by *T. brucei*.

Our data describe critical and previously unappreciated cell types and cell-cell interactions associated with chronic CNS infections. We demonstrate that microglia drive inflammatory and anti-parasitic responses in the CNS, and also provide insights into the transcriptional features border associated macrophages (BAMs). These responses are heterogeneous, with microglia and *Cd14*⁺ monocytes displaying strong pro-inflammatory signatures. Their transcriptional programme is consistent with pro-inflammatory responses expected to be triggered in response to pathogenic challenges during the onset, including the production of cytokines (i.e., *Il1b, Tnf*), chemokines (i.e., *Ccl5, Cxcl10*), and an upregulation of molecules associated with antigen processing and presentation. Based on the differential gene expression analysis over the course of infection, we propose a model in which homeostatic

microglia undergo extensive transcriptional remodelling during infection, leading to the acquisition of an IAMNP phenotype, which coincides with the onset of clinical symptoms. This includes the upregulation of gene programmes involved in other neurodegenerative disorders, such as *Apoe, Aif1, Cst7, Itgax, Tyrobp*, and *Trem2* at the point of infection in which clinical symptoms are detected[31–34]. The transcriptional signatures identified in the IAMNP states are reminiscent of those previously reported in neurodegenerative disorders, thus it is tempting to speculate that this represents a core "pathological" transcriptional module that is triggered in microglia in response to insults, irrespective of the nature of such insults (e.g., parasites versus protein aggregates). In the context of chronic *T. brucei* infection, the acquisition of a IAMNP phenotype might contribute to pathogen clearance or the timely removal of dying/dead parasites, but it is currently unclear whether these subsets are detrimental or beneficial to limit brain pathology. Notably, the expression of putative genes associated with the recognition of pathogen-associated molecular patterns, such as Toll-like receptors and Dectins, was restricted to specific subsets (e.g., *Cd14*⁺ monocytes). Also, the ontogeny of the myeloid subsets exclusively identified during infection remains to be fully elucidated. Although we propose that the two "infection-associated" myeloid subsets identified are likely to be microglia based on their similarities to previously reported disease-associated microglia[62], we cannot exclude the possibility that they might also be derived from perivascular or peripheral myeloid cells that engraft in the brain in response to chronic inflammation. It is therefore tempting to speculate that the "priming" of the myeloid subsets towards an IAMNP state could be triggered by soluble inflammatory mediators such as cytokines and chemokines (e.g., *Ifng, Ccl5, Il1b, Il18*) instead of a direct contact with *T. brucei*. Additionally, the upregulation of genes associated with antigen presentation suggests an active crosstalk with infiltrating T cells, as recently discussed in other neurodegenerative disorders and infections[63–65], but whether the interactions between different subsets are detrimental or beneficial to limit brain pathology remains to be fully elucidated. Additionally, our datasets suggest that *Mrc1*⁺ BAMs acquire an anti-inflammatory state in the chronically infected brain, which might counterbalance the inflammatory responses of IAMNP. This is consistent with the responses observed in other organs in which macrophages with an anti-inflammatory phenotype act to limit inflammatory damage by promoting tissue repair[66–69]. To our knowledge, this is the first report describing the responses of the BAMs to infection by protozoan pathogens. Given their seemingly important role in promoting anti-inflammatory responses, further work is required to explore whether these BAMs consists of ependymal (epiplexus or supraependymal macrophages) or stromal choroid plexus macrophages, as recently discussed[70].

This study also improves our understanding of the components of the adaptive immune response that are recruited into the hypothalamic and brain parenchyma during chronic infection. These include T cells, consistent with previous findings that these cells have a

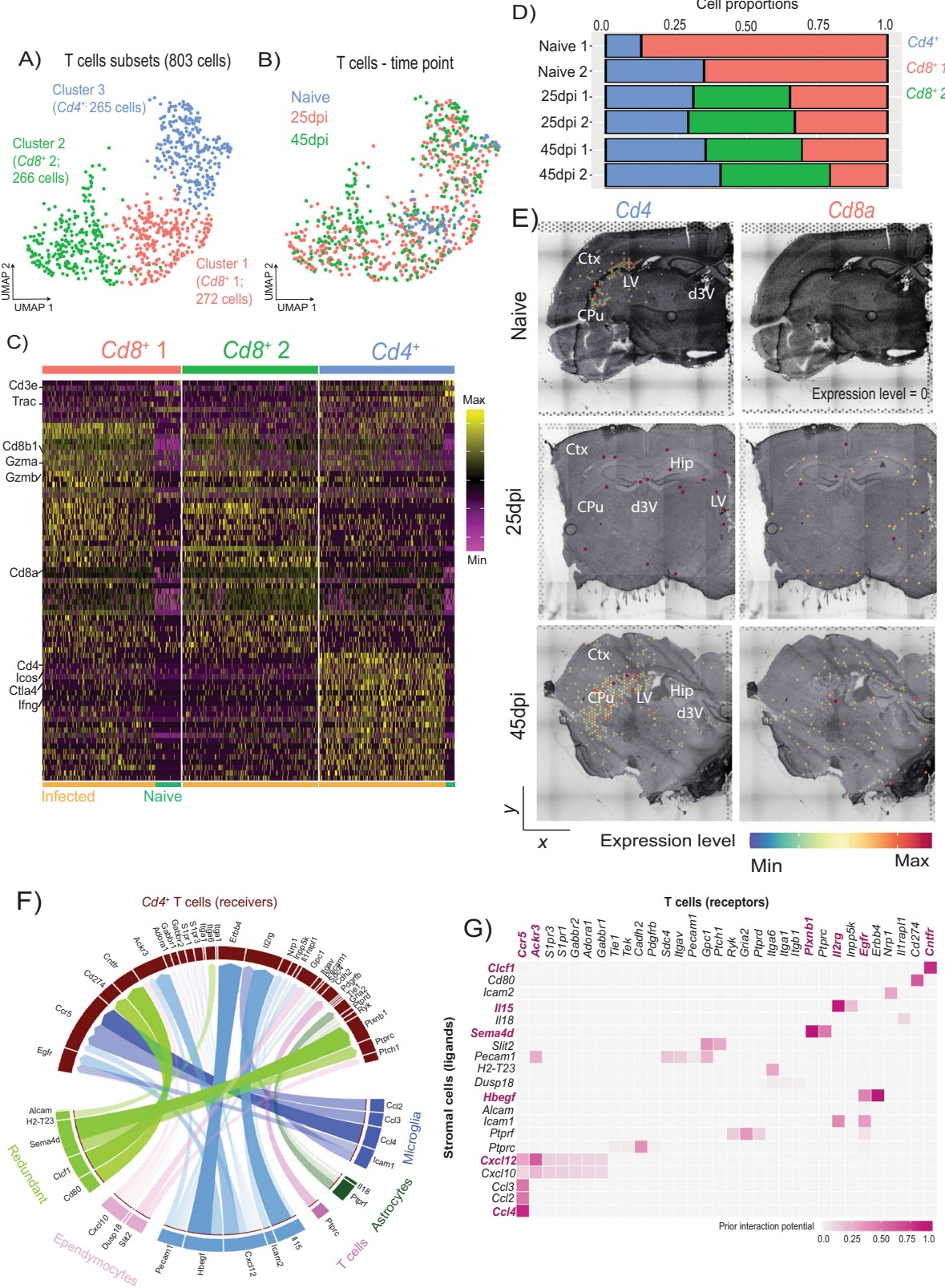

prominent role in modulating CNS responses to *T. brucei* infection[5,7,71]. In addition to conventional cytotoxic *Cd8*+ T cells, we have identified a *Cd4*+ T cell subset that shares many features with T follicular helper cells, including the expression of immunomodulatory genes and effector molecules. We also identified and validated, for the first time, a population of *Cd138*+ plasma cells that display a marked regulatory phenotype, characterised by the expression of *Il10*, *Lgals1*, and *Cd274*. These plasma cells are exclusively detected in chronic infections but not in healthy controls or during the onset of the CNS stage, suggesting a positive correlation between CNS invasion and plasma cell recruitment. Using an in vitro approach, we also show that these cells dampen pro-inflammatory responses in microglia-mediated by IL-10

**Fig. 4 | Chronic *T. brucei* infection leads to an expansion of resident follicular-like *Cd4*⁺ T cells in the CNS. A** UMAP plot depicting the three main T cell subclusters identified in Fig. 1, including the total number of cells in this group. **B** As in **A** but depicting the identified different time points including in this study. **C** Heatmap representing the expression level of putative microglia marker genes for each T cell subcluster. The cell origin within each cluster (Naïve in teal, infected in orange) is also indicated. **D** Proportion of the main T cell subclusters identified in Fig. 1E over the course of infection with *T. brucei*. **E** Spatial feature plot depicting the expression of *Cd4* and *Cd8a*, putative marker genes for *Cd4*⁺ and *Cd8*⁺ T cells, respectively, from samples harvested from naïve (top) and 25dpi (middle), and

45dpi (bottom) coronal mouse brain sections. **F** Circos plot representing significant cell-cell interactions mediated by ligand-receptor communication between T cells (CD4⁺ and CD8⁺ T cells) and microglia (dark blue), astrocytes (dark green), T cells (dark pink), endothelial cells (light blue), and ependymocytes (light pink). Ligand-receptor interactions that were redundantly observed in more than one cell type are shown in light green. **G** Heatmap representing the most significant ligand-receptor interactions between T cells and stromal cells detected in the scRNAseq datasets. The ligand-receptor interaction probability is scored based on the prior interaction potential scale. Ctx cerebral cortex, CPu caudoputamen, Hip Hippocampus, Lateral ventricle (LV); Dorsal 3ʳᵈ ventricle (d3V).

signalling, consistent with previous studies[72–74], although additional factors (e.g., *Lgals1*) might also play a role in this process. In silico predictions suggest that microglia, in particular homeostatic microglia, are able to promote plasma cell survival via *Tnfsf13*, encoding the B cell survival factor BAFF[75–78]. We validated this using smFISH, confirming that both *Tnfsf13* and its cognate receptor *Tnfrsf17* (encoding for the B cell maturation antigen, BCMA) are upregulated in microglia and B cells, respectively, upon infection.

Based on these results, we propose a two-phase model to explain the behaviour of microglia and adaptive immune cells over the course of CNS infection by *T. brucei*, centrally coordinated at the CVOs. The first phase, or "priming phase" takes place during the onset of the CNS stage and is associated with an upregulation of genes involved in antigen presentation, cell migration and response to chemokine signalling, potentially derived from peripheral and/or systemic signals. The second stage, or the "pathology phase", and is characterised by the upregulation of a core transcriptional programme previously reported in neurodegenerative disorders, including *Apoe* and *MHC-II*[31,79–82]. This coincides with the worsening of the clinical scoring and the appearance of severe neurological symptoms in this model of infection. In this context, the activation of homeostatic microglia leads to the recruitment and survival of plasma cells mediated by BAFF-BCMA signalling. In turn, these regulatory plasma cells alleviate inflammation by dampening microglia activation *via* IL-10 signalling, limiting pathology, providing novel insights into the mechanisms of B cell-microglia interactions in the brain during infection. Our model is in line with previous work demonstrating that systemic IL-10 administration ameliorates neuroinflammation during chronic *T. brucei* infection[83], thus highlighting a prominent role of IL-10, derived from either *Cd138*⁺ plasma cells or follicular-like *Cd4*⁺ T cells, in limiting brain pathology. Future work is required to understand the origin of the *Cd138*⁺ plasma cells recruited into the brain parenchyma upon CNS colonisation, but the meningeal space and the lymphopoietic niche at the CNS border is a plausible candidate[56,84]. Our results also demonstrate that the CNS invasion by African trypanosomes is orchestrated and fine-tuned by a myriad of cellular interactions between resident stromal cells and recruited peripheral immune cells in and around the CVOs, suggesting a previously unappreciated role for the CVOs in the pathogenicity of Sleeping Sickness. Notably, the presence of various cell types, including macrophages with tissue remodelling capacity (e.g., *Mrc1*⁺ macrophages), follicular-like *Cd4*⁺ T cells, and plasma cells, resembles the formation of reticular networks typically found in secondary and tertiary lymphoid aggregates. Additionally, we detect a robust expression of genes associated with the formation of lymphoid aggregates such as *Cxcl13*, *Cxcl10*, *Ltb*, and *Tnfsf13b*, which is similar to those recently reported in neuropsychiatric lupus[85]. Thus, it is tempting to speculate that chronic *T. brucei* infection leads to the formation of reticular networks resembling tertiary lymphoid aggregates, supported by follicular-like *Cd4*⁺ T cells, together with stromal cells that might function to support T-B cell interactions (e.g., ependymal cells), ultimately supporting primary humoral responses. We suggest that the *Cd138*⁺ plasma cells identified in our study facilitate this response, especially around the CVOs. However, further work is required to determine whether these structures indeed exist in the chronically

infected brain, and the individual contribution of the various cell types identified in this study to brain pathogenesis and circadian disruptions in sleeping sickness.

Although our work represents a valuable gene expression resource of the murine CNS in response to infection, validated by complementary approaches, further work on examining the expression of other key molecular markers may offer additional information in the field. Similarly, we have also defined several key cell-cell communications taking place in the infected brain using in silico ligand-receptor interaction analysis, but detailed functional experiments are required to validate their role in vitro and in vivo. For instance, the origin of the brain-resident plasma cells, observed in the dorsal 3ʳᵈ ventricle under homeostatic conditions, remains to be evaluated. Additionally, the clonality and antibody repertoire of brain-dwelling plasma cells, expanded after the onset of the CNS stage, merits further investigation. From the parasite perspective, although we did not capture enough parasites in our single-cell transcriptomics datasets to make statistical inferences, we have resolved the spatial distribution of slender and stumpy developmental stages and have provided insights into the transcriptional signatures of these developmental stages in different parts of the murine forebrain, which has remained elusive. Future sorting strategies to purify tissue-dwelling parasites will be greatly beneficial to overcome these challenges. We envision that integration of our work with future scRNAseq, and spatial transcriptomic datasets will address some of the questions arising from this study.

## Methods

### Cell lines and in vitro culture

Murine microglia cell line BV2 (kindly gifted by Dr. Marieke Pingen, University of Glasgow) were cultured in DMEM medium (Sigma) supplemented with 10% foetal bovine serum (FBS) (Sigma) and 1000 IU/mL penicillin, and 100 mg/ml streptomycin. Cells were maintained at 37 °C and 5% $CO_2$. All of the experiments presented in this study were conducted with cells between passages 3 to 6. We challenged BV2 cells with *Escherichia coli* B55:O5 LPS (Sigma) for a period of 2 h to trigger an initial pro-inflammatory response, and then incubated these cells with either untreated B cell supernatant, or B cell supernatant pre-treated with a recombinant antibody to deplete IL-10 (IgG2b, clone JES5-16E3, Biolegend). As controls, BV2 cells were left untreated or were incubated with *E. coli* LPS.

### Gene expression analysis by qRT-PCR

Total RNA from BV2 murine microglia cell lines was extracted using RNeasy Kit (Qiagen), eluted in 30 µl of nuclease-free water (Qiagen), and quantified using Qubit broad range RNA assay (Invitrogen). qRT-PCR analysis was carried out using the Luna Universal One-Step RT-qPCR kit (NEB) using 100 ng RNA as input according to the manufacturer's protocol, using the primers listed below. For each sample, two technical replicates were included, as well as a nuclease-free water sample as a "no template sample" control to determine background signal. The relative expression was calculated using the $2^{-\Delta\Delta Ct}$ formula, where ΔΔCt represents the normalized Ct value of the target RNA relative to the 18 S rRNA and compared to naïve controls. Statistical analysis was conducted using the Mann-Whitney test and *p*-values

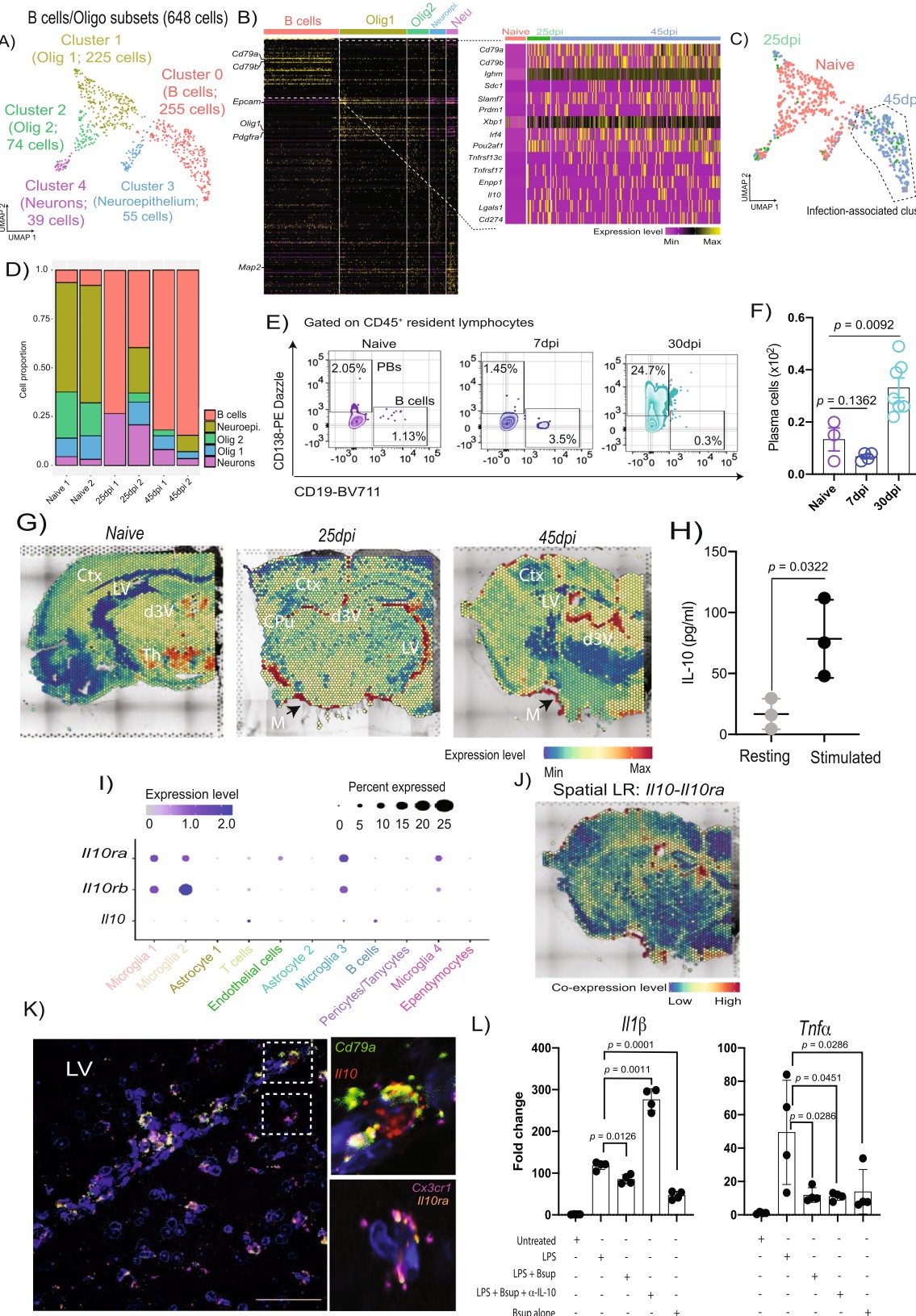

<0.05 were considered statistically significant. The list of primers used in this study is reported in Supplementary Table 1.

**Murine infections with *Trypanosoma brucei***

Six- to eight-week-old female C57Black/6J mice (JAX, stock 000664) were inoculated by intra-peritoneal injection with ~2 × 10³ parasites of strain *T. brucei brucei* Antat 1.1E[86]. Parasitaemia was monitored by regular sampling from tail venesection and examined using phase microscopy and the rapid "matching" method[87]. Uninfected mice of the same strain, sex and age served as uninfected controls. Mice were fed *ad libitum* and kept on a 12 h light–dark cycle. All the experiments were conducted between 8 h and 12 h. For sample collection, we

**Fig. 5 | Regulatory *Cd138*+ plasma cells expand during chronic *T. brucei* infection.** **A** UMAP plot of oligodendrocyte/B cell subclusters. **B** Heatmap of the top 50 most abundant genes in the subcluster in (A). **C** As in **A** but depicting the identified different time points. Dotted line represents clusters exclusively detected in infected samples. **D** Cell proportion over the course of infection. **E** Flow cytometry of CD138+ plasma cells and CD19+ B cells in naïve ($n = 3$ mice), the first peak of infection (7dpi; $n = 4$ mice) and after the onset of the CNS stage (30 dpi; $n = 7$ mice) using flow cytometry. **F** Total number of plasma cells detected by flow cytometry, and representative from two independent experiments. A *p* value of <0.05 is considered significant and was determined using a parametric two-sided *T* test. Data are presented as mean values+/− SD. **G** In silico projection of *Cd138*+ plasma cells distribution onto the spatial transcriptomics slides from naïve (left), 25dpi (middle), and 45dpi (right) coronal mouse brain sections. The relative expression level of is colour coded. **H** IL-10 production by ex vivo brain-dwelling B cells measured by ELISA ($n = 3$ animals per group, repeated twice independently). An adjusted *p*-value of <0.05 is considered significant and was determined using a parametric two-sided *T* test. Data are presented as mean values+/− SD. **I** Expression level of *Il10ra*, *Il10rb*, and *Il10* in the cell types identified in our datasets. **J** Predicted ligand-receptor interaction analysis for *Il10-Il10ra* in the mouse brain at 45dpi. The relative expression level is indicated, and colour coded. **K** smFISH targeting *Il10 (red), Il10ra (orange), Cd79a (green)*, and *Cx3cr1 (purple)* around the LV in the infected mouse brain, including an inlet section, highlighting the co-expression of the predicted ligand-receptor pairs. Scale bar, 25 μm. The results presented here are representative from two independent experiments. **L** qRT-PCR analysis of inflammatory cytokines in BV2 microglia exposed to LPS in the presence of the B cell supernatant ($n = 4$ mice) with or without an anti-IL-10 blocking antibody. Data are presented as mean values+/− SD. Pairwise comparisons were conducted using a two-sided Mann-Whitney test with Welsh correction. *P* values <0.05 is considered significant. *$p < 0.05$; **$p < 0.005$; ***$p < 0.0005$. d3V dorsal 3rd ventricle, Th Thalamus, Ctx Cortex, CPu Caudoputamen, LV Lateral ventricle.

focussed on the onset of the CNS stage (25 days post-infection) and the onset of neurological symptoms (>30 days post-infection), defined in this study as altered gait, reduced co-ordination of hind limbs, and flaccid and/or intermittent paralysis in at least one hind limb. The clinical scoring system to assess disease progression was as follows: score (0) normal, healthy, and explorative mouse; score (1) slow, sluggish, or displaying stary coat; score (2) animals with reduced coordination of hind limbs and/or altered gait; score (3) animals with flaccid paralysis of one hind limb. Animals displaying higher clinical scores (muscle atrophy, complete paralysis, or moribund) were humanely killed immediately in accordance with ethical regulations in our animal project license.

**Brain slice preparation for hypothalamus single-cell RNA sequencing**

**Tissue processing and preparation of single cell suspension.** Single-cell dissociations for scRNAseq experiments were performed as follow. Animals were infected for 25 and 45 days ($n = 2$ mice / time point), after which hypothalami were harvested for preparation of single cell suspensions. Uninfected animals were also included as naive controls ($n = 2$ mice). Briefly, all mice were killed by rapid decapitation following isoflurane anaesthesia, within the same time (between 8:00 and 10:00 AM). Using a rodent brain slicer matrix (Zivic Instrument), we generated ~150 μm coronal brain sections around the hypothalamic area (bregma − 1.34 mm to −1.82 mm, including anterior and posterior hypothalamic structures). The hypothalami were then rapidly excised under a dissection microscope, and the excised hypothalami were then enzyme-treated for ~30 min at 37 °C using protease XXIII (2.5 mg/ml; Sigma) and DNAse I (1 mg/ml; Sigma) in Hank's Balanced Salt Solution (HSBB) (Invitrogen). Slices were washed three times with cold dissociation solution then transferred to a trypsin inhibitor/bovine serum albumin (BSA) solution (1 mg/ml. Sigma) in cold HBSS (Invitrogen). Single-cell suspensions were passed through 70 μm nylon mesh filters to remove any cell aggregates, diluted to ~1,000 cells/μl (in 1X phosphate buffered saline (PBS) supplemented with 0.04% BSA), and kept on ice until single-cell capture using. In parallel, a fraction of these samples was analysed by flow cytometry to estimate the relative proportion of various glial cell types in the single cell suspensions (S1A Figure).

The single cell suspensions were loaded onto independent single channels of a Chromium Controller (10X Genomics) single-cell platform. Briefly, ~20,000 single cells were loaded for capture using 10X Chromium NextGEM Single cell 3 Reagent kit v3.1 (10X Genomics). Following capture and lysis, complementary DNA was synthesized and amplified (12 cycles) as per the manufacturer's protocol (10X Genomics). The final library preparation was carried out as recommended by the manufacturer with a total of 14 cycles of amplification. The amplified cDNA was used as input to construct an Illumina sequencing library and sequenced on a Novaseq 6000 sequencers by Glasgow polyomics.

**Read mapping, data processing, and integration.** For FASTQ generation and alignments, Illumina basecall files (*.bcl) were converted to FASTQs using bcl2fastq. Gene counts were generated using Cellranger v.6.0.0 pipeline against a combined *Mus musculus* (mm10) and *Trypanosoma brucei* (TREU927) transcriptome reference. After alignment, reads were grouped based on barcode sequences and demultiplexed using the Unique Molecular Identifiers (UMIs). The mouse-specific digital expression matrices (DEMs) from all six samples were processed using the R (v4.1.0) package Seurat v4.1.0[88]. Additional packages used for scRNAseq analysis included dplyr v1.0.7[89], RColorBrewer v1.1.2 (http://colorbrewer.org), ggplot v3.3.5[90], and sctransform v0.3.3[91]. We initially captured 25,852 cells mapping specifically against the *M. musculus* genome across all conditions and biological replicates, with an average of 37,324 reads/cell and a median of ~615 genes/cell (Supplementary Data 14). The number of UMIs was then counted for each gene in each cell to generate the digital expression matrix (DEM) (Supplementary Fig. 1B). Low quality cells were identified according to the following criteria and filtered out: (i) nFeature <200 or >1500, (ii) nCounts <200 or >5,000, (iii) >10% reads mapping to mitochondrial genes, and (iv) >40% reads mapping to ribosomal genes, (v) genes detected <3 cells. After applying this cut-off, we obtained a total of 13,195 high quality mouse-specific cells with an average of 12,162 reads/cells and a median of 577 genes/cell (Supplementary Data 14).

We noted that the overall number of UMIs was significantly higher in samples from 25 and 45dpi compared to naïve controls (Supplementary Fig. 1B). A closer examination of the number of genes/UMIs per cell type enabled us to determine that the overall increase in infected samples derived mostly from microglia and Oligodendrocytes/B cells (Supplementary Fig. 1B) and may be indicative of a "transcriptional burst" associated with cell activation. Based on these observations, we considered this differential feature and gene counts when scaling the data (see below). The gene counts for each cell were divided by the total gene counts for the cell and multiplied by the scale factor 10,000. Then, natural-log transformation was applied to the counts. To identify gene signatures that represent highly variable genes (HVGs) we employed two independent approaches: (i) The Seurat *FindVariableFeatures* function with default parameters, using *vst* as selection method, and (ii) The *plotHighestExprs* in *Scater* package[92] with default parameters, which allowed us to manually inspect the HVGs detected by these methods (Supplementary Fig. 1C). We then applied the Seurat function *SCTransform* for data normalisation, scaling, and variance stabilisation of HVGs, regressing out for percentage of mitochondrial and ribosomal genes, total UMIs, genes counts, and cell cycle genes. This was followed by data integration using *IntegrateData* and *FindIntegrationAnchors*. For this, the number

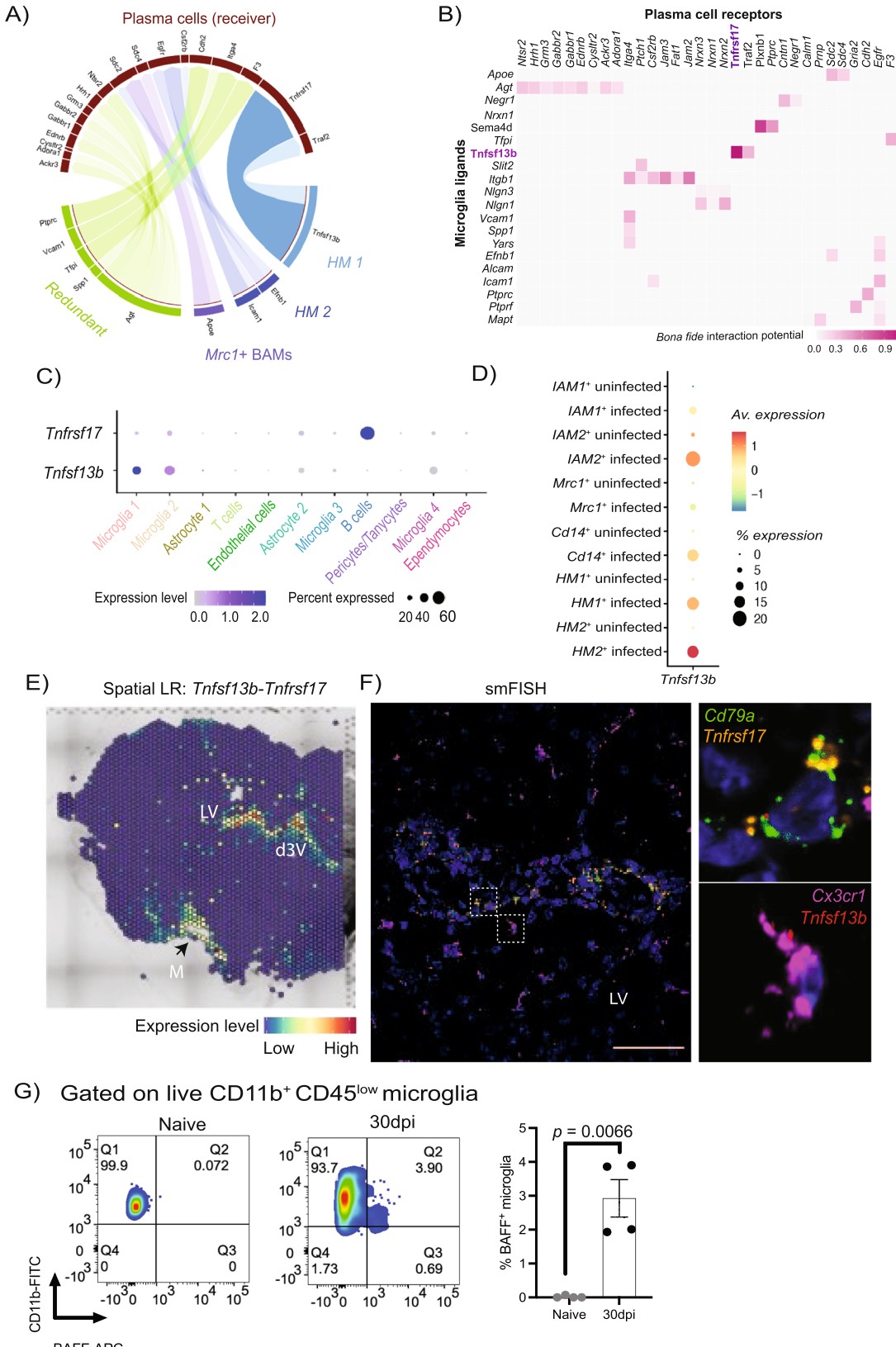

of principal components were chosen using the elbow point in a plot ranking principal components and the percentage of variance explained (10 dimensions) using a total of 5000 genes, and SCT as normalisation method. In parallel, given the gene/UMIs discrepancies between experimental groups, we analysed the integrated dataset using the STACAS workflow[93] with default parameters (10 dimensions)

to determine if the clusters identified with the Seurat package can be reproduced by an independent method. Overall, we detected the same marker genes and cell types identified by the *IntegrateData* and *FindIntegrationAnchors* function in Seurat. We applied the same approach with the myeloid subset with similar results, suggesting that the differential gene/UMI counts between experimental groups (and

**Fig. 6 | Crosstalk between *Cd138*⁺ plasma cells and homeostatic microglia in the brain of chronically infected animals. A** Circos plot of significant ligand-receptor interactions between *Cd138*⁺ plasma cells and HM 1 (light blue), HM 2 (dark blue), and *Mrc1*⁺ BAMs (purple). Redundant interactions (shared by >1 cell type) are shown in light green. **B** Heatmap of the most significant ligand-receptor interactions between *Cd138*⁺ plasma cells and microglia. **C** Dot Plot representing the expression level of *Tnfsf13b* (BAFF) and its cognate receptor *Tnfrsf17* (BCMA). **D** Dot Plot representing the expression level of *Tnfsf13b* (BAFF) in the various microglia sub-sets identified in the mouse hypothalamus during *T. brucei* infection. **E** Predicted spatial ligand-receptor interaction analysis for *Tnfsf13b-Tnfrsf17* in the mouse brain at 45dpi. The relative expression level is indicated, and colour coded. **F** Chronic *T.* *brucei* infection induces the expression of *Tnfsf13b* and *Tnrsf17* in homeostatic microglia and B cells, respectively. Representative smFISH probe targeting *Cx3cr1* (purple), *Cd79a* (green), *Tnfsf13b* (red), and *Tnfrsf17* (orange) around the lateral ventricle (LV) in an infected mouse brain coronal section. *Cx3cr1* was chosen as a marker for homeostatic microglia. Scale, 25 μm. The results presented here are representative from two independent experiments. **G** Representative flow cyto-metry analysis and quantification of BAFF⁺ microglia in naïve and infected animals (30 dpi) using flow cytometry (n = 3 naïve or 4 infected mice from a single repli-cate). Data are presented as mean values+/− SD. Pairwise comparison was con-ducted against naïve samples using a parametric T test. $p < 0.05$ is considered significant. **$p < 0.005$.

accounted for when scaling the data) does not cofound downstream detection of marker genes or cell types.

**Cluster analysis, marker gene identification, and subclustering.** The integrated dataset was then analysed using *RunUMAP* (10 dimensions), followed by *FindNeighbors* (10 dimensions, reduction = "pca") and *FindClusters* (resolution = 0.4). With this approach, we identified a total of 11 cell clusters The cluster markers were then found using the *FindAllMarkers* function (logfc.threshold = 0.25, assay = "RNA"). To iden-tify cell identity confidently, we employed a hierarchical approach, combining unsupervised and supervised cell identity methods. For the unsupervised approach, we implemented two complementary R packages, scCATCH[94] and SingleR[95], using hypothalamic datasets built-in as references with default parameters (Supplementary Data 3). These packages map a query dataset against selected reference atlases, scoring the level of confidence for cell annotation. Cell identities were assigned based on confidence scores and/or independent cell identity assignment by the two packages. Confidence scores >85% were assumed to be reliable and the cell annotations were kept. When the two packages failed to detect cell identity confidently (confidence scores <85% by at least one package), we employed a supervised approach. This required the manual inspection of the marker gene list followed by and assignment of cell identity based on the expression of putative marker genes expressed in the unidentified clusters. This was particularly relevant for immune cells detected in our dataset that were not found in the reference atlases used for mapping. A cluster name denoted by a single marker gene indicates that the chosen candidate gene is selectively and robustly expressed by a single cell cluster and is sufficient to define that cluster (e.g., *Cd79a, Cd8a, C1qa, Cldn5,* among others). The addition of a second marker was used to indicate a sec-ondary identifier that is also strongly expressed in the cluster but shared by two or more subclusters (e.g., *Apoe, Gfap*).

When manually inspecting the gene markers for the final cell types identified in our dataset, we noted the co-occurrence of genes that could discriminate two or more cell types (e.g., macrophages from microglia). To increase the resolution of our clusters to help resolve potential mixed cell populations embedded within a single cluster and, we subset vascular associated cells (endothelial cells, pericytes, tanycytes, and ependymocytes), microglia, T cells, and oligodendrocytes (the latter also containing a distinctive B cell cluster) and analysed them individually using the same functions described above. In all cases, upon subsetting, the resulting objects were reprocessed using the functions *FindVariableFeatures, RunUMAP, FindNeighbors*, and *FindClusters* with default parameters. The number of dimensions used in each cased varied depending on the cell type being analysed but ranged between 5 and 10 dimensions. Cell type-level differential expression analysis between experimental conditions was conducted using the *Findallmarkers* function (*min.pct* = 0.25, *test.use* = Wilcox) and (*DefaultAssay* = "SCT"). Cell-cell interaction analysis mediated by ligand-receptor expression level was conducted using NicheNet[43] with default parameters using "mouse" as a reference organism, comparing differentially expressed genes between experimental conditions (*condition_oi* = "Infected",

*condition_reference* = "Uninfected"). Pathways analysis for mouse genes were conducted using STRING[96] with default parameters.

Module scoring for inflammatory mediators were calculated using the *AddModuleScore* function to assign scores to groups of genes of interest (*Ctrl* = 100, *seed* = NULL, *pool* = NULL), and the scores were then represented in violin plots. This tool measures the average expression levels of a set of genes, subtracted by the average expres-sion of randomly selected control genes. The gene list was collated from the integrated scRNAseq Seurat object using the function grep for known pro- and anti-inflammatory cytokines and chemokines. Once defined, the collated gene list was used to build the module scoring. Statistical tests using the non-parametric Wilcox test com-paring mean of normalised gene expression (basemean) was conducted in R.

⁴⁴⁹⁸Raw data and scripts used for data analysis will be made pub-licly available after peer review.

**10X Visium spatial sequencing library preparation and analysis Tissue processing and library preparation.** Coronal brain sections (bregma−1.34 mm to −1.82 mm) were frozen in optimal cutting tem-perature medium (OCT) and stored at −80 °C until sectioning. Opti-mization of tissue permeabilization was performed on 10-μm-thick sections using the Visium Spatial Tissue Optimization Reagents Kit (10X Genomics), which established an optimal permeabilization time of 18 min. Samples were mounted onto a Gene Expression slide (10X Genomics) and stored at −80 °C until haematoxylin and eosin (H&E) staining. To prepare for staining, the slide was placed on a thermo-cycler adaptor set at 37 °C for 5 min followed by fixation in ice-cold methanol for 30 min. Methanol was displaced with isopropanol and the samples were air-dried for 5-10 min before sequential staining with Mayer's haematoxylin Solution (Sigma-Aldrich), Bluing Buffer (Dako) and 1:10 dilution of Eosin Y solution (Sigma-Aldrich) in 0.45 M of Tris-acetic acid buffer, pH 6.0, with thorough washing in ultrapure water between each step. Stained slides were scanned under a microscope (EVOS M5000, Thermo). Tissue permeabilization was performed to release the poly-A mRNA for capture by the poly(dT) primers that were precoated on the slide, including a spatial barcode and a Unique Molecular Identifiers (UMIs). The Visium Spatial Gene Expression Reagent Kit (10X Genomics) was used for reverse transcription and second strand synthesis, followed by denaturation, to allow the transfer of the cDNA from the slide to a collection tube. These cDNA fragments were then used to construct spatially barcoded Illumina-compatible libraries using the dual Index Kit TT Set A (10x Genomics) was used to add unique i7 and i5 sample indexes, enabling the spatial and UMI barcoding. The final Illumina-compatible sequencing library was sequenced on a single lane (2 × 150) of a NextSeq 550 instrument (Illumina) by Glasgow Polyomics.

After sequencing, the FASTQ files were aligned to a merged reference transcriptome combining the *Mus musculus* genome (mm10) genome and the *Trypanosome brucei* reference genome (TREU927). We have found that this approach leads to a better gene identification in host-pathogen dual transcriptomics experiments. When mapping against the *T. brucei* transcriptome alone, we identified

that -0.7%, 1.2%, and 1.4% of the total reads map to the *T. brucei* transcriptome, with a median of 9, 7, and 67 *T. brucei*-specific genes per spot in naïve, 25dpi, and 45 dpi samples, respectively. After alignment using the merged reference transcriptome, reads were grouped based on spatial barcode sequences and demultiplexed using the UMIs, using the SpaceRanger pipeline version 1.2.2 (10X Genomics). Downstream analyses of the expression matrices were conducted using the Seurat pipeline for spatial RNA integration[88,97] (Supplementary Data 15), and the overall gene density per spot, as a quality control metric, in the different tissue sections is reported in Supplementary Fig. 3A. Specifically, the data was scaled using the *SCTransform* function with default parameters. We then proceeded with dimensionality reduction and clustering analysis using *RunPCA* (*assay* = "SCT"), *FindNeighbours* and *FindClusters* functions with default settings and a total of 30 dimensions. We then applied the *FindSpatiallyVariables* function to identify spatially variable genes, using the top 1,000 most variable genes and "markvariogram" as selection method. The approach enabled us to identify 13–19 distinct, spatially resolved transcriptional clusters in the different tissue sections included in this study. We optimised the parameters to obtain clustering of distinct spatially variable gene sets (Supplementary Fig. 3A) that broadly coincide with several brain regions, including cortex, hippocampus, 3$^{rd}$ and lateral ventricles, thalamus, hypothalamus, striatum, and amygdala (Supplementary Fig. 3C), confirming the robustness, reproducibility, and reliability of our data. For the analysis of the *T. brucei* genes detected in the spatial transcriptomics dataset, we used the *SpatialFeaturePlot* function (*alpha* = 0.01, 0.1, *min.cutoff* = 0.1). The genes detected in the spatial transcriptomics dataset at 45dpi where further analysed using the gene ontology server built in the TriTrypDB website[98] with default settings. Module scoring for *T. brucei* genes were calculated using the *AddModuleScore* function to assign scores to groups of genes of interest (*Ctrl* = 100, *seed* = NULL, *pool* = NULL), and the scores were then represented in violin plots. Once defined, the collated gene list was used to build the module scoring. Statistical tests using the non-parametric Wilcox test comparing mean of normalised gene expression (basemean) was conducted in R.

To integrate our hypothalamic scRNAseq with the 10X Visium dataset, we used the *FindTransferAnchors* function with default parameters, using SCT as normalization method. Then, the *TransferData* function (*weight.reduction* = "pca", 30 dimensions) was used to annotate brain regions based on transferred anchors from the scRNAseq reference datasets. To predict the cell-cell communication mediated by ligand-receptor co-expression patterns in the spatial context, we employed NICHES v0.0.2[99]. Upon dimensionality reduction and data normalisation, NICHES was run using fanton5 as ligand-receptor database with default parameters. The resulting object was then scaled using the functions *ScaleData, FindVariableFeatures* (*selection.method* = "disp"), *RunUMAP* with default settings and a total of 15 dimensions. Spatially resolved expression of ligand-receptor pairs was then identified using the *FindAllMarkers* function (*min.pct* = 0.25, *test.use* = "roc"). For visualisation, we used the *SpatialFeaturePlot* function with default parameters and *min.cutoff* = "q1". Raw data and scripts used for data analysis will be made publicly available after peer review.

**Immunofluorescence and single molecule fluorescence in situ hybridisation (smFISH) using RNAscope.** Formalin-fixed paraffin embedded coronal brain sections were section on a microtome (Thermo) and fixed in 4% PFA for 10 min at room temperature. Sections were blocked with blocking buffer (1X PBS supplemented with 5% foetal calf serum and 0.2% Tween 20) and incubated with primary antibodies (Supplementary Table 2) at 4 °C overnight, followed by incubation with fluorescently conjugated secondary antibodies for 1 h at room temperature. All the antibodies were diluted in blocking buffer. Slides were mounted with Vectashield mounting medium containing DAPI for nuclear labelling (Vector Laboratories) and were

visualized using an Axio Imager 2 (Zeiss). The list of antibodies for immunofluorescence and RNAscope probes used in this study is presented in the table below.

smFISH experiments were conducted as follow. Briefly, to prepare tissue sections for smFISH, infected animals and naïve controls were anesthetized with isoflurane, decapitated and brains were dissected out into ice-cold 1X HBSS. Coronal brain sections were prepared as described above and embedded in paraffin. Cryopreserved coronal brain sections (5 μm) were prepared placed on a SuperFrost Plus microscope slides. Sections were fixed with 4% paraformaldehyde (PFA) at 4 °C for 15 min, and then dehydrated in 50, 70, and 100% ethanol. RNAscope 2.5 Assay (Advanced Cell Diagnostics) was used for all smFISH experiments according to the manufacturer's protocols. All RNAscope smFISH probes (Supplementary Table 3) were designed and validated by Advanced Cell Diagnostics. For image acquisition, 16-bit laser scanning confocal images were acquired with a 63x/1.4 plan-apochromat objective using an LSM 710 confocal microscope fitted with a 32-channel spectral detector (Carl Zeiss). Lasers of 405 nm, 488 nm, and 633 nm excited all fluorophores simultaneously with corresponding beam splitters of 405 nm and 488/561/633 nm in the light path. 9.7 nm binned images with a pixel size of 0.07 um × 0.07 um were captured using the 32-channel spectral array in Lambda mode. Single fluorophore reference images were acquired for each fluorophore and the reference spectra were employed to unmix the multiplex images using the Zeiss online fingerprinting mode. smFISH images were acquired with minor contrast adjustments as needed, and converted to grayscale, to maintain image consistency. The resulting images were processed and analysed using QuPath[100], and the values plotted using Prism v8.0. The in situ hybridisation images were acquired from the publicly available resource the Allen Mouse Brain Atlas (www.mouse.brain-map.org/) and used in Supplementary Fig. 3B.

**Flow cytometry analysis and ex vivo stimulation of brain-dwelling B cells.** All the antibodies used for flow cytometry are provided in Supplementary Table 4. To discriminate circulating versus brain-resident immune cells, we performed intravascular staining of peripheral CD45$^+$ immune cells as previously reported[101]. Briefly, a total of 2 μg of anti-CD45-PE antibody (in 100 μl of 1X PBS) was injected intravenously 3 min prior culling. Mice were euthanised as described above and transcardially perfused with ice-cold 0.025% (wt/vol) EDTA in 1X PBS. Whole brain samples were collected and placed on ice-cold 1X HBSS (Invitrogen) and processed as recently described[102]. Whole brain specimens were minced and digested using the Adult Brain dissociation kit (Miltenyi) for 30 min at 37 °C, following manufacturer's recommendations. The digested tissue was gently pressed through 70 μm nylon mesh cell strainers to obtain a single cell suspension. The cell suspension was cleaned up and separated from myelin debris using a Percoll gradient, as previously reported[102]. The resulting fraction was then gently harvested and used as input for glia profiling or for B cell purification using the B cell isolation kit II (negative selection approach) using MACS sorting (Miltenyi). Cells from spleens were used as positive controls. The resulting cell fraction was diluted to a final density of -1 × 10$^6$ cells/ml. The resulting suspension enriched in B cells were seeded on a 96-well plate and stimulated with 1X cell Stimulation cocktail containing phorbol 12-myristate 13-acetate (PMA), Ionomycin, and Brefeldin A (eBioSciences$^{TM}$) for 5 h at 37 °C and 5% $CO_2$, as previously reported[55]. Upon stimulation, the resulting supernatant was harvested and used to quantify IL-10 by ELISA (Biolegend), or to test its capacity to block BV2 polarisation in the presence of *E. coli* LPS. As control, anti-mouse IL-10 antibody (IgG2b, clone JES5-16E3. Biolegend) was applied to the B cell-derived supernatant for 30 min at room temperature to sequester and block IL-10 signalling in vitro.

For flow cytometry analysis, single cell suspensions were resuspended in ice-cold FACS buffer (2 mM EDTA, 5 U/ml DNAse I, 25 mM HEPES and 2.5% Foetal calf serum (FCS) in 1X PBS) and stained for

extracellular markers at 1:400 dilution. The list of flow cytometry antibodies used in this study were obtained from Biolegend and are presented in the table below. Samples were run on a flow cytometer LSRFortessa (BD Biosciences) and analysed using FlowJo software version 10 (Treestar). For intracellular staining, single-cell isolates from brain or draining lymph nodes were stimulated as above in Iscove's modified Dulbecco's media (supplemented with 1× non-essential amino acids, 50 U/ml penicillin, 50 μg/ml streptomycin, 50 μM β-mercaptoethanol, 1 mM sodium pyruvate and 10% FBS. Gibco). Cells were then permeabilized with a Foxp3/Transcription Factor Staining Buffer Set (eBioscience) and stained for 30 min at 4 °C. All antibodies used were diluted at 1:250.

### Reporting summary

Further information on research design is available in the Nature Research Reporting Summary linked to this article.

## Data availability

The data generated in this study have been deposited in the Gene Expression Omnibus database under accession code GSE200642. The processed transcript count data and cell metadata generated in this study are available at Zenodo (https://zenodo.org/record/6387555#.YwkaFi8w1nk)[103]. The flow cytometry data generated in this study are provided in the Supplementary Information/Source Data file. Additional data and files can also be sourced via Supplementary Datas. Source data are provided with this paper. The single cell dataset can be explored in this link: https://cellatlas-cxg.mvls.gla.ac.uk/tbrucei_brain/. Source data are provided with this paper.

## Code availability

Code used to perform analysis described can be accessed at Zenodo (https://zenodo.org/record/6387555#.YkW3tC8w1nk)[34].

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

## Acknowledgements

We thank Julie Galbraith and Pawel Herzyk (Glasgow Polyomics, University of Glasgow) for their support with library preparation and sequencing. Similarly, we would like to thank the technical staff at the University of Glasgow Biological Services for their assistance in maintaining optimal husbandry conditions and comfort for the animals used in this study, and the Flow Core Facility (School of Infection and Immunity) for technical support. We thank Dr. Jean Rodgers for the work conducted under her Home Office Animal License (PPL no. PC8C3B25C). We thank Dr. Sara Macias Ribela, Dr. Virginia Howick, and Dr. Paul Capewell for critically reading this manuscript and for providing feedback. The authors would also like to thank the III Flow cytometry facility for their support with flow cytometry analysis. This work was funded by a Sir Henry Wellcome postdoctoral fellowship (221640/Z/20/Z to J.F.Q.) and an Wellcome Trust ISSF Catalyst grant awarded to J.F.Q. (204820/Z/16/Z to JFQ). EMB is funded by a Sir Henry Wellcome postdoctoral fellowship (218648/Z/19/Z to E.M.B.). AML is a Wellcome Trust Senior Research fellow (209511/Z/17/Z to A.M.L.). P.C. and M.C.S. are supported by a Wellcome Trust Senior Research fellowship (209511/Z/17/Z to A.M.L.) awarded to A.M.L. N.A.M. is supported by project Institute Strategic Programme Grant funding from the BBSRC (BBS/E/D/20231762 and BBS/E/D/20002174 to N.A.M.). T.D.O. is supported by a Wellcome Trust grant to Andrew Waters (104111/Z/14/ZR).

## Author contributions

Methodology: J.F.Q., A.M.L., N.A.M., L.D.L. In vivo work and sample collection: J.F.Q., M.C.S. Bioinformatic data analysis (single cell transcriptomics): J.F.Q., P.C., E.M.B., T.D.O. Bioinformatic data analysis (spatial transcriptomics): J.F.Q. Flow cytometry: J.F.Q., M.C.S. Imaging: J.F.Q., P.C., R.H., C.B.A., C.L. The single-cell atlas was created by T.D.O. All authors participated in discussions and interpretations of the results presented in this work. J.F.Q wrote the manuscript. All authors reviewed and approved the manuscript.

## Competing interests

The authors declare no competing interests.

## Ethics

All animal experiments were approved by the University of Glasgow Ethical Review Committee and performed in accordance with the home office guidelines, UK Animals (Scientific Procedures) Act, 1986 and EU

directive 2010/63/EU. All experiments were conducted under SAPO regulations and UK Home Office project licence number PC8C3B25C to Dr. Jean Rodger. The in vivo work related to the single cell and spatial transcriptomic experiments were conducted at 25- and 45-days post-infection (dpi) and correlated with increased clinical scores and procedural severity. Subsequent in vivo experiments for experimental validation (flow cytometry and imaging) were terminated earlier in line with ethical recommendations from the veterinary team at the University of Glasgow.
