## [Peer Review File · Nature Communications]

Single cell and spatial transcriptomic analyses reveal microglia-plasma cell crosstalk in the brain during *Trypanosoma brucei* infectionREVIEWER COMMENTS

Reviewer #2 (Remarks to the Author):

The manuscript from Quintana et al utilizes single cell transcriptomics, including spatial transcriptomics) to examine the CNS response to *T. brucei* infection. The authors identify that circumventricular organs appear to have the most parasite and corresponding inflammation. The transcriptomic analyses indicate several flavors of microglia/monocytes in the CNS including those with high expression of microglia homeostatic markers and those consistent with activated microglia and monocyte-derived cells. The authors also identify that B cells are a source of IL-10 and microglia are producing BAFF, a survival factor for B cells. The manuscript is likely to be an incredible resource for those studying the biology of chronic *T. brucei* infection of the brain.

1) For the transcriptomic analysis, my most major comment is on the use of genes like *Aif1* and *CX3CR1* to define the microglia/monocyte/macrophage clusters. While these genes are highly expressed, they are certainly not unique to microglia. Thus, it would be helpful to possibly rename the clusters to reduce potential confusion regarding the identity of the clusters. Perhaps the clusters could be called based on genes like *tmem119*, *sall1* that are unique to microglia (even if genes like *tmem119* are downregulated with inflammation). The text states that cells in some of the clusters are expressing microglia homeostatic genes but at a lower level, which is a more powerful way to define these clusters versus *Aif1*-expression. Given the high number of CD14-expressing monocytes, it is reasonable to conclude that these cells may differentiate into cells that acquire markers like *cx3cr1*, MHC II, and *Aif1* in particular.

2) Could comment on whether there is evidence of tertiary lymphoid structures forming that would further support B cells in the CNS during infection. Do they preferentially form in the circumventricular organs?

3) The identification of a "disease-associated" microglia population is very interesting, including whether the DAM program is beneficial or detrimental. The authors state that the DAM signature appears when pathology increases, but this may or may not be connected to microglia. The authors should speculate on whether MHC or Dectin-1 (or any others in the DAM signature) are likely to be involved in tissue destruction or a brain-protective immune response. It's a timely topic and worthy of discussion.

Minor comments:

1) *CCL2* is typically associated with the recruitment of *CCR2*-expressing monocytes and not neutrophils (typically *CXCR1* and *CXCR2* for PMNs).

2) BAMs also express *CX3CR1*... What other than *Arg1* makes them anti-inflammatory. They are typically defined by *CD206* expression, but what else might make them functionally anti-inflammatory?

Reviewer #3 (Remarks to the Author):

The authors have studied the transcriptional response in the brain-hypothalamus to *T. brucei* using single cell RNA-seq and spatial transcriptomics.

The methods are really well described and with the data analysis scripts being made available upon publication the work should be reproducible.

- The authors have used 10x v3.1 to generate single cell RNA-seq data. Though they are targeting the immune cells of the brain which tend to have lower gene expression diversity during tissue homeostasis than, for example, neurons have, the median number of genes detected per cell is really low. The cutoff used was 200 genes per cell and in the naive condition the median nr of genes per cell is 270, just barely above the cutoff. The low gene detection levels should be (and are according to the figures) good enough to distinguish major cell types, but when looking at the transcriptional response to a perturbation a lot of information is lost and bias is introduced. Looking at the data, the quality could probably have been improved by sequencing deeper. Could the authors comment?

- In panel 3G the authors show the result of a DEG analysis in the 6 different microglial clusters between Naive and 25/45dpi. The cell populations, naive, 25dpi, and 45dpi are very different in terms of gene detection level. Seeing this many genes differentially regulated (more than the median gene detection in the cells) indicates that the genes that appear in the DEG list are not robustly expressed across most cells. Furthermore, the comparison within the Aif1/2 types between naive and 25/45dpi is very unbalanced in terms of cells from each condition included in that clusters. The Aif1/2 types hardly contain any naive cells. Both the gene detection level and the cell numbers in each population can lead to overinterpretation of the result.

- There are some problems with the tissue used for visium analysis. The brain sections are cracked and some show significant folding. Both these issues cause an over- or underrepresentation of genes in the visium spot. The visium data was aligned to both M musculus and T brucei genomes. Pathway analysis was performed on T brucei transcript but there is no mention of how many genes per spot were detected for that genome. I would assume that gene detection levels are lower than for the mouse genome. How meaningful than is the pathway analysis just on those genes?

Minor:

- Figure 3F, remove the dashed lines to figure 3E. Panel F is not taken from that area in panel E. It would be better to include a representative atlas image.

- The supplemental tables are hard to read without formatting and Table S3C is too small to read.

-

Reviewer #4 (Remarks to the Author):

Previous work has demonstrated that *T. brucei* infection results in glial cell activation and neuroinflammation. However, the molecular mechanisms that govern these processes are unknown. This work utilizes two powerful approaches, scRNAseq and spatial transcriptomics, to define the transcriptional responses of different classes of immune cells within a well-defined area of the brain at two different time points in a rodent model of *T. brucei* infection.

This work is of interest to researchers from diverse fields including molecular parasitologists, immunologists, and bioinformaticians that are working to integrate multiple, distinct, large-scale 'omics data sets to understand dynamic responses. Overall, I found the study to be well designed and informative. This reviewer agrees that the work represents a useful resource for studying cellular and molecular events that occur during brain infection and provides a model for how the integration of single cell RNA seq and spatial transcriptomics can be used to study host-parasite interactions.

There were several instances where the transcripts mentioned in the text were not listed in the figure and cell clusters mislabeled. It would be important to carefully proofread all figures and text for any discrepancies. I have noted some below in the major and minor points, but they are not exhaustive and there may be additional differences I haven't detected.

Questions/comments on text:

- Lines 204-209 regarding the identification of outliers. Is it possible to document genes that were flagged as "outliers". Might it be useful to other researchers to know the identity of these outliers? Also, do genes have to be identified by both approaches before being designed as outliers.? Can the authors provide a reference for "highly variable genes such as long non-coding RNAs such as Malat1?

- Table "List of RNAscope probes". To what is "channel" referring? Would it be better to indicate wavelength and/or detector?

- Lines 431-432. How do the 500 genes/cell and 1500 transcript/cell compare to other studies? Is this higher or lower than expected?

- Line 449- in reference to figure 1G- How was the inflammatory module score calculated?
- Line 532- should Figure 3F be Figure 3E?
- Line 607-I could not find Icam1 in the Figure 4F graph
- Line 622, I could not find Sox10 and S100b in figure 1D
- Lines 625-628: There percentages of each cell cluster given in the text does not agree with the image in figure 5A.
- Lines 1025, Figure 5 legend. There is no description of panel L.
- There are several spatial transcriptome figures including Figure 2A, Figure 3E, and Figure 5E that are described differently. Can the authors use consistent descriptors or highlight the differences between the experiments.

Questions/comments on Figures:

- Figure 1A). The right side of the figure panel is not clear. This can likely be solved by more information in the figure legend detailing the significance of the different colors.
- Figure 1C) What are the parameters used to determine clinical score and how are they quantified?
- Figure 1D) How are inflammatory gene module scores determined? I couldn't not find a reference for "in silico gene module" score
- Figure 1E) I interpreted the heat map to be comparisons among the cell lines in the figure and not between infected and naïve samples. Is that correct?
- Figure 2) I found this figure particularly compelling, and some questions came to mind. Were any brain sections stained for BOTH stumpy and slender markers simultaneously? I am curious whether there might be a mixture of slender and stumpy parasites in different regions. Is there a way to compare the number of parasites present with the amount of transcript detected in the spatial transcripts? For example, is it possible to discriminate between a few, very actively transcribing parasites from more, less actively transcribing parasites? Please provide more information for the left-hand Venn diagram in E. Was the GO Term analysis done only with the 969 transcripts that were found exclusively in the CVOs or for all 1067 transcripts?
- Figure 3: There is a discrepancy regarding significance levels. Figure legend (lines 947-948) indicates " $-2 < \text{Log}_2\text{Fold change} < 2$ adjusted p value < 0.05 " while the text (lines 544-545) indicates " $\text{Log}_2\text{fold change} > 0.25$ or < -0.25 ".
- Figure 3B: It is unclear why some of the transcripts are in bold.
- Figure 3E: Justification for using Adgre1 and Chil3 is not provided.
- Figure 4: Lines 587-591 indicates a 2-fold increase in abundance that is not apparent in the cell proportions indicated in Figure 4D. Also the numbers of each subcluster indicated in the text in lines 589-593 do not match with the cell proportions given in the table in Figure 4D.
- Figure 5 A. Should Cluster 5 be Cluster 0? The percentages given in the text for the clusters in A do not match what is provided in the figure panel 5A. For example, Cluster 1 = 34% (line 625) and Cluster 3 = 8.48% (lines 626). In the figure, there are fewer dots in cluster 1 than 3. There is no description for Panel L.
- Figure 6. How are the average and percent expression in panel D calculated?

- Figure S1. More information is needed to interpret the flow cytometry panels. Are “infected” samples from 25 or 45 dpi? Macs and Micros are not defined. The live dead assay is not provided. The gating strategy in C is not clear from the figure legend. There is not enough information in the legend to interpret panel C. Tnfsf13b and Tnfrsf17 are not shown.

- Figure S2. Could the transcript names be color coded to indicate which cluster they are associated?

- Figure S3. It is unclear to me what the quality controls experiments are testing: the integrity of the brain regions during processes or the reproducibility in the number of transcriptional clusters detected? What should the reader expect to see if the data pass quality control? Also, it is unclear to me what the 18 clusters represent. Are they the same clusters as in Figures 1 and S5?

**Decision on Nature Communications manuscript NCOMMS-22-13214**

We thank all the reviewers for their positive assessments and helpful comments. We
have summarised the changes made to the manuscript in response, and we believe
that they have significantly improved it.

Major points:

- 1. We have re-annotated the myeloid subsets identified in our single cell dataset
to better reflect their potential function (e.g., *Cd14*⁺ monocytes, homeostatic
microglia, infection-associated microglia) as suggested by reviewer 2.
2. We have commented on the potential role of the newly identified infection-
associated microglia subsets in the context of chronic infection and speculate
how these cells might be interacting with peripheral immune cells to
coordinate anti-parasitic responses and during brain pathology, as
recommended by reviewer 2.
3. We have addressed the observation of low gene counts per non-neuronal
cells at a steady state level and have commented why we believe this might
be the case, as raised by reviewer 3. Moreover, we have compared the
dataset presented in this manuscript against other previously published
articles describing the use of similar technologies to study the murine
hypothalamus and/or murine glial cells, as queried by reviewer 4.
4. We have commented on the overall quality of the tissue sections used for
spatial transcriptomics, as highlighted by reviewer 4. Both single cell and
spatial transcriptomics were used in this study as a starting point to identify
novel interactions in the context of chronic brain infections, and the finding
derived from these *in silico* predictions were independently validated by
several methods throughout the manuscript (e.g., imaging, flow cytometry),
further validating our bioinformatic predictions.
5. We have created a searchable database that will enable the community to
explore and analyse the single cell data reported in this manuscript:
https://cellatlas-cxg.mvls.gla.ac.uk/tbrucei_brain/

Below we have provided point-by-point answers to the major and minor comments
raised by these three reviewers.

**REVIEWER COMMENTS**

**Reviewer #2 (Remarks to the Author):**

The manuscript from Quintana et al utilizes single cell transcriptomics, including
spatial transcriptomics) to examine the CNS response to T. brucei infection. The
authors identify that circumventricular organs appear to have the most parasite and
corresponding inflammation. The transcriptomic analyses indicate several flavors of
microglia/monocytes in the CNS including those with high expression of microglia
homeostatic markers and those consistent with activated microglia and monocyte-
derived cells. The authors also identify that B cells are a source of IL-10 and
microglia are producing BAFF, a survival factor for B cells. The manuscript is likely to
be an incredible resource for those studying the biology of chronic T. brucei infection
of the brain.

We appreciate such positive feedback from this reviewer and their enthusiasm and
support for our findings.

1) For the transcriptomic analysis, my most major comment is on the use of genes
like Aif1 and CX3CR1 to define the microglia/monocyte/macrophage clusters. While
these genes are highly expressed, they are certainly not unique to microglia. Thus, it
would be helpful to possibly rename the clusters to reduce potential confusion
regarding the identity of the clusters. Perhaps the clusters could be called based on
genes like tmem119, sall1 that are unique to microglia (even if genes like tmem119
are downregulated with inflammation). The text states that cells in some of the
clusters are expressing microglia homeostatic genes but at a lower level, which is a
more powerful way to define these clusters versus Aif1-expression. Given the high
number of CD14-expressing monocytes, it is reasonable to conclude that these cells
may differentiate into cells that acquire markers like cx3cr1, MHC II, and Aif1 in
particular.

We thank the reviewer for their insights. As this reviewer states, clusters 0 and 1
display high expression levels of genes used to identify “homeostatic” microglia such
as *Tmem119*, *Sall1*, *Cx3cr1*, *Hexb*, *P2ry12*, amongst others. Clusters 4 and 5 do not
show detectable expression levels of homeostatic microglia-related genes, but they
do display a robust transcriptional programme consistent with disease-associated
microglia (DAM) previously reported in neurodegenerative conditions. These include
*Aif1*, *Cst3*, *Cst7*, as well as genes associated with phagocytosis and lipid metabolism
such as *Apoe*, *Ctsd*, and *Tyrobp*, amongst others (**reported in table S2F**). Though
we agree that some of these genes can be shared by infiltrating myeloid cells that
acquire a microglia-like phenotype in the infected brain milieu, as recently reported
for myeloid cells during spinal cord injury in mice¹, we cannot decipher the ontogeny
of these cells with the current dataset with enough certainty. We have renamed the
*Aif1*⁺ clusters as “Infection-associated microglia” (IAM) and the *Cx3cr1*⁺ clusters as
“homeostatic microglia” throughout the text to better reflect the underlying complexity
of these myeloid subsets. These changes are also reflected in the updated figure 3,
where we have also included the marker genes for each cluster in panel 3A.

We have changed the following text and associated figures to reflect these changes
as follow:

**Line 634:** *Based on these results we catalogued clusters 0 to 5 as follow: Homeostatic*
*microglia (HM) 1 (1,688 cells; 26.77%), HM 2, (1,548 cells; 24.55%), Cd14⁺ Monocytes*
*(1,396 cells; 22.14%), Mrc1⁺ border-associated macrophages (Mrc1⁺ BAMs – 812*

cells; 12.6%), Infection-associated microglia (IAM) 1 (587 cells; 9.31%), and IAM 2
(274 cells; 4.34%) (Figure 3C and D). Notably, HM 1 and 2, and Cd14⁺ monocytes
accounted for ~73% of all the microglia detected under homeostatic conditions, but
IAM 1 and 2, and Mrc1⁺ BAMs subclusters progressively increased in frequency over
the course of infection, suggesting an adoption of an infection-associated phenotype
(Figure 3C and 3D).

2) Could comment on whether there is evidence of tertiary lymphoid structures forming
that would further support B cells in the CNS during infection. Do they preferentially
form in the circumventricular organs?

This is a really interesting question. The formation of tertiary lymphoid structures in
the CNS was recently reported during chronic neuroinflammatory and autoimmune
conditions. Based on our own data, we detect genes associated with lymphoid
structures such as lymphotoxin b (*Ltb*), *Cxcl13*, and *Tnfsf13b*, among others. We are
currently investigating this in more detail.

We have added a comment on the discussion to refer to this, as follow:

**Line 1131:** *Notably, the presence of various cell types, including macrophages with*
*tissue remodelling capacity (e.g., Mrc1⁺ macrophages), follicular-like CD4⁺ T cells,*
*and plasma cells, resembles the formation of reticular networks typically found in*
*secondary and tertiary lymphoid aggregates. Additionally, we detect a robust*
*expression of genes associated with the formation of lymphoid aggregates such as*
*Cxcl13, Cxcl10, Ltb, and Tnfsf13b, which is similar to those recently reported in*
*neuropsychiatric lupus*². *Thus, it is tempting to speculate that chronic T. brucei*
*infection leads to the formation of reticular networks resembling tertiary lymphoid*
*aggregates, supported by follicular-like CD4⁺ T cells, together with stromal cells that*
*might function to support T-B cell interactions (e.g., ependymal cells), ultimately*
*supporting primary humoral responses. We suggest that the Cd138⁺ plasma cells*
*identified in our study facilitate this response, especially around the CVOs. However,*
*further work is required to determine whether these structures indeed exist in the*
*chronically infected brain, and the individual contribution of the various cell types*
*identified in this study to brain pathogenesis and circadian disruptions in sleeping*
*sickness.*

3) The identification of a “disease-associated” microglia population is very
interesting, including whether the DAM program is beneficial or detrimental. The
authors state that the DAM signature appears when pathology increases, but this
may or may not be connected to microglia. The authors should speculate on whether
MHC or Dectin-1 (or any others in the DAM signature) are likely to be involved in
tissue destruction or a brain-protective immune response. It’s a timely topic and
worthy of discussion.

We do not observe demyelination or extensive neurotoxicity at this point of infection
using this model of infection. Similarly, we do not observe the overexpression of
*Clec7a* (encoding for Dectin-1), and *Tlr2* was only identified in the Cd14⁺ monocyte
subset. We think it is unlikely that any of the responses reported in this manuscript
are associated with tissue destruction, beyond potentially tissue remodelling around
the ventricular spaces and the meninges, consistent with ventricular and meningeal
inflammation. It remains to be explored if the overexpression of MHC leads to the
activation of other adaptive immune cells also detected in the brain parenchyma

during infection, such as CD4⁺ and CD8⁺ T cells. As suggested, we have added text
to speculate about this:

**Lines 1029:** *In the context of chronic T. brucei infection, the acquisition of a IAM*
*phenotype might contribute to pathogen clearance or the timely removal of dying/dead*
*parasites, but it is currently unclear whether these subsets are detrimental or beneficial*
*to limit brain pathology. Notably, the expression of putative genes associated with the*
*recognition of pathogen-associated molecular patterns, such as Toll-like receptors and*
*Dectins, was restricted to specific subsets (e.g., Cd14⁺ monocytes). Thus, it is*
*tempting to speculate that the “priming” of the myeloid subsets towards an IAM state*
*could be triggered by soluble inflammatory mediators such as cytokines and*
*chemokines (e.g., lfn3, Ccl5, Il1b, Il18) instead of direct contact with T. brucei.*
*Additionally, the upregulation of genes associated with antigen presentation suggests*
*an active crosstalk with infiltrating T cells, as recently discussed in other*
*neurodegenerative disorders and infections³⁻⁵, but whether the interactions between*
*different subsets are detrimental or beneficial to limit brain pathology remains to be*
*fully elucidated.*

Minor comments:

1) CCL2 is typically associated with the recruitment of CCR2-expressing monocytes
and not neutrophils (typically CXCR1 and CXCR2 for PMNs).

We thank the reviewer for this comment. We have now amended this in line 728.

2) BAMs also express CX3CR1... What other than Arg1 makes them anti-
inflammatory. They are typically defined by CD206 expression, but what else might
make them functionally anti-inflammatory?

Myeloid-specific cluster 3 express high levels of *Mrc1*, which encodes for CD206, in
addition to additional markers traditionally associated with anti-inflammatory
macrophages. Though this cell population highly expresses several putative anti-
inflammatory macrophage markers, we did not detect a cytokine profile typically
associated with this function (e.g., *Il10* or *Il4*). Rather, we detected expression of *Il1b*
(pro-inflammatory properties) and *Il18bp* (anti-inflammatory properties), which may
indicate either heterogeneous populations within this cluster, or the expression of
mixed cytokines. Nevertheless, we cannot resolve these nuances with the current
dataset, and we are now working towards better resolving these populations *in vitro*
and *in vivo*. We have also amended the text as follows:

**Line 623:** *Cluster 3 expresses putative marker genes associated with border-*
*associated macrophages such as *Lyz2*, *Ms4a7*, *Ms4a6c*, *Tgfbi*, *H2-Ab1*, and *Lyz2**
*^{6,7}, as well as gene sets characteristic of anti-inflammatory responses, such as *Mrc1**
*(encoding for CD206), *Chil3*, *Arg1*, and *Vegfa* (Figure 3B and S2F Table), indicative*
*of an anti-inflammatory phenotype.*

**Reviewer #3 (Remarks to the Author):**

The authors have studied the transcriptional response in the brain-hypothalamus to
193 T. brucei using single cell RNA-seq and spatial transcriptomics.

The methods are really well described and with the data analysis scripts being made
available upon publication the work should be reproducible.

- The authors have used 10x v3.1 to generate single cell RNA-seq data. Though they
are targeting the immune cells of the brain which tend to have lower gene
expression diversity during tissue homeostasis than, for example, neurons have, the

median number of genes detected per cell is really low. The cutoff used was 200
 genes per cell and in the naive condition the median nr of genes per cell is 270, just
 barely above the cutoff. The low gene detection levels should be (and are according
 to the figures) good enough to distinguish major cell types, but when looking at the
 transcriptional response to a perturbation a lot of information is lost and bias is
 introduced. Looking at the data, the quality could probably have been improved by
 sequencing deeper. Could the authors comment?

We thank the reviewer for their thoughts on this aspect of the paper and we agree
 regarding the relatively low median number of genes per cell. To address this, we
 have described below some points which reassured us that the data generated are
 of good quality despite the relative low gene counts per cell in the naïve controls:

a. During our optimisation steps, we sequenced two biological replicates in a
 pilot 10X experiment and consistently detected low number of genes per cell
 from hypothalamic preparations in naïve samples (Figure 1A). Moreover, the
 complexity score, which should be >0.8 ⁸, was higher in the samples included
 in this study compared to the ones from the pilot dataset (Figure 1B).

**Figure 1. Comparison of the QC metrics between the naïve samples included in this study**
 **and from a pilot experiment.** A) Number of genes detected per cell. The dotted lines
 represents the cut-off applied to these samples. In all cases, after filtering based on the
 number of genes per cell, we recovered 75-90% of the input cells. B) Complexity of the dataset, or
 novelty score, for the samples in the current and pilot study. All these samples display a score >0.8 .

b. In all cases, the percentage of live cells was consistently $>85\%$ as determined
 by flow cytometry, ruling out potential issues associated with dying/dead cells.
 **We have added the corresponding % of viability per sample in table S2C.**
 c. The samples presented in this manuscript were sequenced at a saturation
 $>92\%$, and so we did not try to increase the depth of sequencing any further
 as we assumed an additional $\sim 5\%$ sequencing would not resolve this
 apparent low gene counts drastically. **We have added the corresponding %**
 **sequencing saturation per sample in table S2C.**
 232 d. As the reviewer indicates, our bioinformatic pipeline captured the major non-
 233 neuronal cell populations that we would expect to see in the hypothalamus
 based on previously reported single cell atlases⁹⁻¹³.
 e. Given the gene discrepancies between experimental groups, we analysed the
 integrated dataset using two independent computational approaches (Seurat
 and STACAS) and detected the same marker genes discussed in this

manuscript. Notably, the number of genes per cell differs between cell types
under homeostatic conditions. We draw this reviewer's attention to Figure
S1B (upper panel), where we reported that the B cells/oligodendrocytes
cluster has, on average, twice as many genes per cells than ependymocytes,
for example, despite uniform number of UMI per cell (**Figure S1B lower**
**panel**). This might indicate heterogeneity in the overall gene detection level
across cell type within the same sample.

f. We consistently detected greater median gene number per cell in infected
samples compared to naïve samples processed in parallel.

It is worth noting that the reported genes and UMIs per cells vary greatly across
reports, with some recent studies reporting "low gene counts" for glial cells in both
murine and human brain tissues ranging from 400-800 genes/cell and ~1,500-3,000
UMIs/cell^{10,11,13-15, 1}. Though these are important parameters to understand the
underlying biology of these cells, we have found that these are inconsistently
reported in the currently available literature. **We have included further information**
**in the revised table S2A** to increase transparency in the results presented in this
manuscript.

- In panel 3G the authors show the result of a DEG analysis in the 6 different
microglial clusters between Naive and 25/45dpi. The cell populations, naive, 25dpi,
and 45dpi are very different in terms of gene detection level. Seeing this many genes
differentially regulated (more than the median gene detection in the cells) indicates
that the genes that appear in the DEG list are not robustly expressed across most
cells.

Broadly, we analyse gene signatures that are present in at least 25% of the cells, as
defined in the *Findallmarkers* function in Seurat. In the context of an experimental
infection, we would expect a strong upregulation of genes not transcribed under
homeostatic conditions. For instance, several of the pathways reported in figure 3H
(e.g., antigen processing and presentation, chemotaxis, etc) are typically found in
activated myeloid cells¹⁶⁻¹⁸¹⁹. Thus, the list of significantly dysregulated gene in
response to infection is likely to reflect a true biological response to a perturbation, in
this case, infection. We draw your attention to **line 805** where we stated the
following: *Most of the upregulated DEGs were detected in the HM 1, HM 2, and*
*Cd14⁺ monocyte subclusters (Figure 3G).*

Furthermore, the comparison within the Aif1/2 types between naive and 25/45dpi is
very unbalanced in terms of cells from each condition included in that clusters. The
Aif1/2 types hardly contain any naive cells. Both the gene detection level and the cell
numbers in each population can lead to overinterpretation of the result.

We agree with this reviewer that the proportion of Aif1/2 types between naïve and
infected samples is unbalanced. To avoid overinterpretation, we have removed this
comparison from figure 3G. We have now included an additional figure panel (Figure
3I), showing the gene pathways enriched in Aif1/2 types based on their
transcriptional profile (e.g., marker genes). Additionally, we have added additional
text to clarify that Aif1/2 clusters were not included in the pathway analysis:

**Line 815:** *We did not include cells within the IAM subclusters as the overall cell*
*proportion is reduced in naïve compared to infected mice, potentially confounding*
*DEG analysis. Instead, we analysed these two subclusters separately to better*
*understand their transcriptional features. We found that the cells within the IAM1*

*subcluster display the expression of gene pathways broadly associated with antigen*
 *processing and presentation (H2-Ab1, H2-Aa, H2-Eb1), neutrophil activation (Ccl5,*
 *Fcgr4, Fcer1g), and synaptic pruning (C1qc, C1qb, C1qa), whereas those within IAM*
 *2 also upregulate genes associated with translational activity (Rps5, Rps14, Rpsa,*
 *Rps15), suggesting a transcriptionally active state (Figure 3I and Table S2I).*

We have also removed the Aif1/2 clusters from the DEGs list in supplementary table
 S2G and S2H. We have included the complete list of enriched pathways in
 supplementary tables S2I, as well as in the table legend as follow:

**Line 1750:** S2I) List of gene pathways identified in the IAM1 and IAM2 clusters.

Significant pathways are considered those with a false discovery rate (FDR) < 0.05.

- There are some problems with the tissue used for visium analysis. The brain
 sections are cracked and some show significant folding. Both these issues cause an
 over- or underrepresentation of genes in the visium spot.

The overall quality, measured by transcript and gene distribution per spot in the
 array, was consistent across samples. We have included a separate spatial feature
 plot depicting the overall number of genes per spot in supplementary figure 3A.

The visium data was aligned to both M musculus and T brucei genomes. Pathway
 analysis was performed on T brucei transcript but there is no mention of how many
 genes per spot were detected for that genome. I would assume that gene detection
 levels are lower than for the mouse genome. How meaningful than is the pathway
 analysis just on those genes?

As discussed in the methods section (**line 362**), we used a purpose-built reference
 transcriptome combining the *Mus musculus* and the *T. brucei* transcriptome. We
 have found in previous work that this approach results in better gene identification
 and annotation in host-pathogen dual transcriptome studies. To answer this question
 specifically, we have repeated the alignment using only the reference *T. brucei*
 transcriptome and have found that 0.7%, 1.2%, and 1.4% of the total reads map to
 the *T. brucei* transcriptome, with a median of 9, 7, and 67 *T. brucei*-specific genes
 319 per spot in naïve, 25dpi, and 45 dpi samples, respectively. We have included the
 320 results from SpaceRanger in the table below to summarise these observations:

Sample	Mean reads per spot	Median T. brucei genes per spot	Top T. brucei marker genes
Naïve	75,224	9	Tb11.v5.0333, Tb11.v5.0349, Tb11.v5.0444, Tb11.v5.0524, Tb11.v5.0653, Tb11.v5.0701, Tb11.v5.0806, Tb11.v5.0813, Tb11.v5.0852
25dpi	67,575	7	Tb11.v5.0813, Tb11.v5.0349, Tb11.v5.0806, Tb11.v5.0653
45dpi	90,379	67	Tb11.v5.0444, Tb11.v5.0852, Tb11.v5.0852, Tb11.v5.0701, Tb11.v5.0813, Tb11.v5.0333, Tb11.v5.0349, Tb11.v5.0524, Tb927.1.4540, Tb927.1.2390

We have amended the methods section to report these observations as follow:

**Line 364:** We have found that this approach leads to a better gene identification in
 host-pathogen dual transcriptomics experiments. When mapping against the *T.*
 *brucei* alone, we identified that ~0.7%, 1.2%, and 1.4% of the total reads map to the
 *T. brucei* transcriptome, with a median of 9, 7, and 67 *T. brucei*-specific genes per
 spot in naïve, 25dpi, and 45 dpi samples, respectively. After alignment using the
 merged reference transcriptome, reads were grouped based on spatial barcode

*sequences and demultiplexed using the UMIs, using the SpaceRanger pipeline*
*version 1.2.2 (10X Genomics).*

Regarding the pathway analysis, we included this information using the top *T. brucei*-
specific marker genes identified by Seurat to explore potential signatures that define
the brain-dwelling parasites compared to those reported in other tissues/organs,
such as the bloodstream, as this remains poorly understood. Though limited, our
data suggest that the parasites located in the brain ventricles display signatures of
both slender and stumpy developmental forms. Future work is required to explore
this at a finer scale (e.g., using FACS to purify ventricle-enriched parasites).

Minor:

- Figure 3F, remove the dashed lines to figure 3E. Panel F is not taken from that area
in panel E. It would be better to include a representative atlas image.

We have now amended the figure and as suggested have included a representative
atlas image depicting the area from which the images were captured from.

- The supplemental tables are hard to read without formatting and Table S3C is too
small to read.

We thank the reviewer for picking up on this. We have increased the font size in all
the supplementary tables, including Table S3C.

**Reviewer #4 (Remarks to the Author):**

Previous work has demonstrated that *T. brucei* infection results in glial cell activation
and neuroinflammation. However, the molecular mechanisms that govern these
processes are unknown. This work utilizes two powerful approaches, scRNAseq and
spatial transcriptomics, to define the transcriptional responses of different classes of
immune cells within a well-defined area of the brain at two different time points in a
rodent model of *T. brucei* infection.

This work is of interest to researchers from diverse fields including molecular
parasitologists, immunologists, and bioinformaticians that are working to integrate
multiple, distinct, large-scale 'omics data sets to understand dynamic responses.
Overall, I found the study to be well designed and informative. This reviewer agrees
that the work represents a useful resource for studying cellular and molecular events
that occur during brain infection and provides a model for how the integration of
single cell RNA seq and spatial transcriptomics can be used to study host-parasite
interactions.

We thank the reviewer for a positive assessment of the work presented here and the
constructive suggestions, which we have addressed in the revised manuscript as
well as in the sections below:

There were several instances where the transcripts mentioned in the text were not
listed in the figure and cell clusters mislabeled. It would be important to carefully
proofread all figures and text for any discrepancies. I have noted some below in the
major and minor points, but they are not exhaustive and there may be additional
differences I haven't detected.

Questions/comments on text:

•Lines 204-209 regarding the identification of outliers. Is it possible to document

genes that were flagged as “outliers”. Might it be useful to other researchers to know
the identity of these outliers? Also, do genes have to be identified by both
approaches before being designed as outliers.? Can the authors provide a reference
for “highly variable genes such as long non-coding RNAs such as Malat1?”

We thank the reviewer for flagging this. The identification of highly variable genes
(HVGs) is a critical step for the downstream identification of discreet cell populations
385 ^{20–22}. The functions described in the methods section (e.g., using *vst* selection
method in Seurat’s *FindVariableFeatures* function, or *plotHighestExprs* in *Scater*)
allowed us to identify and reduce the impact of technical outliers (e.g., lowly
expressed genes with high dispersion) through variance stabilisation. Additionally,
the tools that identify HVGs are reported to give different results²³, therefore we
employed two independent methods (Seurat and Scater) for internal comparison.
Thus, it is not possible to compute a set number of HVGs as they are likely to vary
depending on the data used as input. Overall, we found that some of the HVGs
identified by Seurat and Scater overlap (e.g., *Malat1*), but we did not require for them
to be identified by both packages for downstream analysis. We have included the top
25 most variable genes identified by Scater in our dataset in **figure S1C** as an
example, and have also amended the text in the methods section to clarify this as
follow:

**Line 226:** *To identify gene signatures that represent highly variable genes (HVGs)*
*we employed two independent approaches: i) The Seurat FindVariableFeatures*
*function with default parameters, using vst as selection method, and ii) The*
*plotHighestExprs in Scater package* ²¹ *with default parameters, which allowed us to*
*manually inspect the HVGs detected by these methods (Figure S1C). We then*
*applied the Seurat function SCTransform for data normalisation, scaling, and*
*variance stabilisation of HVGs, regressing out for percentage of mitochondrial and*
*ribosomal genes, total UMIs, genes counts, and cell cycle genes.*

•Table “List of RNAscope probes”. To what is “channel” referring? Would it be better
to indicate wavelength and/or detector?

The RNAscope probes are provided in different “channels” enabling multiplexing. In
this table we reported the channels chosen for each of the probes. However,
following your question, we have amended this table to include the fluorescent dye
used in each case (**Line 448**).

•Lines 431-432. How do the 500 genes/cell and 1500 transcript/cell compare to other
studies? Is this higher or lower than expected?

This is an important question, also raised by reviewer 3. Surprisingly, there is a lot of
variation in the number of genes and transcripts per cell detected in previous studies
using single cell/nuclei transcriptomics for profiling murine hypothalamus. For
instance, a recent report has implemented similar cut-off as the ones reported in our
study using the hypothalamus from aging female mice ¹⁴, and led to the identification
of similar cell populations as the ones reported here. On a separate report using
human microglia during Alzheimer’s, the authors reported a median 844 genes/cell
and 1,589 UMIs/cell, with some samples reporting as low as ~400 genes/cell ¹⁵. This
is also the case for non-neuronal cells from the murine spinal cord, with a median of
~750 genes/cell¹. However, other reports that profile the transcriptome of neuron and
non-neuron cells in the hypothalamus reported a median of ~2,500 genes/cell and
~6,000 UMIs/cell ^{10–13,25}. These discrepancies might be due to differences in

experimental approaches, or regions profiled within the hypothalamus (e.g., whole
hypothalamus, lateral or posterior hypothalamus, etc.).

•Line 449- in reference to figure 1G- How was the inflammatory module score
calculated?

We first mined the integrated scRNAseq object to identify pro- and anti-inflammatory
cytokines using the function below; We broadly called this compendium of molecules
“cytokine list”:

```
cytokine.list <- c(grep("^Csf", rownames(data), value = T),
grep("^Ifn", rownames(combined_integrated), value = T),
grep("^Il", rownames(combined_integrated), value = T),
grep("^Tnfsf", rownames(combined_integrated), value = T),
grep("^Cxcl", rownames(combined_integrated), value = T),
grep("^Ccl", rownames(combined_integrated), value = T),
"Tslp", "Lif", "Osm", "Tnf", "Lta", "Ltb", "Cd40l", "Fasl",
"Cd70", "Tgfb1", "Mif", "Cx3cl1")
AddModuleScore(combined, features = list(as.character(cytokine.list)))
```

To estimate the enrichment of *T. brucei* transcripts in the spatial transcriptomics
dataset, we employed a similar approach but mining the normalised spatial
transcriptomics for *T. brucei*-specific genes.

The *AddModuleScore* function calculates the average expression levels of each
gene list (in this case, the genes within the cytokine.list set) on a given dataset (e.g.,
single cell of spatial transcriptomics), subtracted by the aggregated expression of
control feature sets randomly selected by the function. This leads to a corrected
expression level for genes of interest, in this case, (pro and anti) inflammatory
cytokines.

We have added the following text in the Methods section of the revised manuscript
describing how this parameter was constructed below, and have made the
corresponding codes available: <https://zenodo.org/record/6387555#.YkW3tC8w1nk>

**Line 309:** *Module scoring for inflammatory mediators were calculated using the*
*AddModuleScore function to assign scores to groups of genes of interest (Ctrl = 100,*
*seed = NULL, pool =NULL), and the scores were then represented in violin plots.*
*This tool measures the average expression levels of a set of genes, subtracted by*
*the average expression of randomly selected control genes. The gene list was*
*collated from the integrated scRNAseq Seurat object using the function grep for*
*known pro- and anti-inflammatory cytokines and chemokines. Once defined, the*
*collated gene list was used to build the module scoring.*

**Line 392:** *Module scoring for T. brucei genes were calculated using the*
*AddModuleScore function to assign scores to groups of genes of interest (Ctrl = 100,*
*seed = NULL, pool =NULL), and the scores were then represented in violin plots. Once*
*defined, the collated gene list was used to build the module scoring. Statistical tests*
*using the non-parametric Wilcox test comparing mean of normalised gene expression*
*(basemean) was conducted in R.*

•Line 532- should Figure 3F be Figure 3E?

We thank the reviewer for spotting this. We have now amended this in 626.

•Line 607-I could not find Icam1 in the Figure 4F graph

We draw the reviewer's attention to the microglia cluster (dark blue; bottom right)
where Icam1 is depicted

•Line 622, I could not find Sox10 and S100b in figure 1D

We thank the reviewer for noticing this oversight and have now added these two
marker genes to the heatmap in figure 1D. We have also referenced table S2J that
should contain all the markers for these clusters as follow:

**Line 933:** *This appeared to represent a heterogeneous grouping of cells expressing*
*high levels of oligodendrocyte markers (Olig1, Sox10, and S100b) and bona fide B*
*cell markers (Cd79a, Cd79b, Ighm) (Figure 1D, S2B and S2J Table).*

•Lines 625-628: There percentages of each cell cluster given in the text does not
agree with the image in figure 5A.

We thank the reviewer for noticing this error. We have now corrected this issue in
figure 5A.

•Lines 1025, Figure 5 legend. There is no description of panel L.

We thank the reviewer for noticing this oversight. We have now added the following
text in the figure caption describing the results presented in panel L:

**Line 1448:** *L) qRT-PCR analysis of pro-inflammatory mediators ($Il1\beta$ and $Tnf\alpha$) in*
*BV2 microglia cell lines exposed to LPS in the presence of the B cell supernatant*
*with our without an anti-IL-10 blocking antibody. Pairwise comparisons were*
*conducted against cells exposed to LPS alone using Mann-Whitney test. A p values*
*<0.05 were considered statistically significant. * $p < 0.05$; ** $p < 0.005$; *** $p <$*
*0.0005.*

•There are several spatial transcriptome figures including Figure 2A, Figure 3E, and
Figure 5E that are described differently. Can the authors use consistent descriptors
or highlight the differences between the experiments.

As requested, we have now amended the captions in Figure 2A (**line 1303**) and
Figure 5G (**line 1435**). Please note that the panels in figure 5 have changed, but that
I have updated the one mentioned here.

Questions/comments on Figures:

•Figure 1A). The right side of the figure panel is not clear. This can likely be solved
by more information in the figure legend detailing the significance of the different
colors.

Thank you for this suggestion. We have amended figure 1A to reflect these proposed
changes.

•Figure 1C) What are the parameters used to determine clinical score and how are
they quantified?

The clinical score is assessed in accordance with our Home Office animal project
license (No. PC8C3B25C). We have included the following statement in the Method
section to clarify the clinical scoring system:

**Line 159:** *The clinical scoring system to assess disease progression was as follow:*
*score 0) normal, healthy, and explorative mouse; score 1) slow, sluggish, or displaying*
*stary coat; score 2) animals with reduced coordination of hind limbs and/or altered*
*gait; score 3) animals with flaccid paralysis of one hind limb. Animals displaying higher*
*clinical scores (muscle atrophy, complete paralysis, or moribund) were humanely killed*
*immediately in accordance with ethical regulations in our animal project license.*

•Figure 1D) How are inflammatory gene module scores determined? I couldn't not
find a reference for "in silico gene module" score

Thank you for this question. We have now addressed this as outlined above (**line**
**309**).

•Figure 1E) I interpreted the heat map to be comparisons among the cell lines in the
figure and not between infected and naïve samples. Is that correct?

Yes, this is correct. The heatmap shows the expression level of the top 25 marker
genes for each of the cell clusters identified in the hypothalamus. The heatmap is
organised based on abundance of each of the clusters. For instance, cluster
"Microglia 1" (far left) has comparatively more cells than the "Ependymocytes" cluster
(far right). We have clarified this in the figure legend as follow:

**Line 1269:** *E) Heatmap representing the expression level of the top 25 marker*
*genes for each of the cell clusters identified in the combined dataset.*

•Figure 2) I found this figure particularly compelling, and some questions came to
mind. Were any brain sections stained for BOTH stumpy and slender markers
simultaneously? I am curious whether there might be a mixture of slender and
stumpy parasites in different regions.

We thank the reviewer for this question. Indeed, we stained the samples presented
in figure 2D with both stumpy and slender markers but did not include the composite
image in the original submission for simplicity. However, we have now included the
composite in this figure and have amended the legend accordingly. Though we
cannot confidently assign spatial segregation of the different forms, we do tend to
see slender forms more scattered across the ventricles and surrounding tissue,
whereas the stumpy forms are mostly confined to the choroid plexus.

Is there a way to compare the number of parasites present with the amount of
transcript detected in the spatial transcripts? For example, is it possible to
discriminate between a few, very actively transcribing parasites from more, less
actively transcribing parasites?

This is a great question. Unfortunately, we cannot address this question as the 10X
Visium data is not resolved at a single cell level. Rather, we are only able to resolve
transcriptional spots, in which there might be an enrichment of "slender" transcripts,
"stumpy" transcripts, or a mixture of both. Future experiments sorting parasites from
various brain regions and comparing them using single cell transcriptomics would
allow us to resolve these populations in greater detail.

Please provide more information for the left-hand Venn diagram in E. Was the GO
Term analysis done only with the 969 transcripts that were found exclusively in the
CVOs or for all 1067 transcripts?

We have amended the figure legend to provide more details. For simplicity, the GO
term analysis was conducted on the 969 (~90%) transcripts detected in the CVOs.

We draw the reviewer's attention to **line 1287**: *E) Ven diagram of the different T.*
*brucei-specific transcripts detected in several brain regions at 45dpi, based on the*
*spatial distribution shown in 4B. Top 10 GO terms that characterise brain-dwelling*
*African trypanosomes located in the CVOs.*

•Figure 3: There is a discrepancy regarding significance levels. Figure legend (lines
947-948) indicates “-2 < Log2Fold change < 2 adjusted p value < 0.05” while the text
(lines 544-545) indicates “Log2fold change > 0.25 or < -0. 25”.

We thank the reviewer for noticing this error. We have amended this in **line 1324** as
follow: *These genes were defined as having a -0.25 < Log₂ Fold change < 0.25, and*
*an adjusted p value of < 0.05*

•Figure 3B: It is unclear why some of the transcripts are in bold.

We thank the reviewer for noticing this and we have amended this in the figure

•Figure 3E: Justification for using Adgre1 and Chil3 is not provided.

We thank the reviewer for this comment. We draw their attention to the text in **line**
**777**: *Similarly, Arg1 and Chil3, putative marker genes for Mrc1⁺ BAMs, were*
*predominantly located in the lateral ventricle and the dorsal 3rd ventricle in the*
*infected brain (Figure 3E), further corroborated by immunofluorescence analysis on*
*independent brain sections (Figure 3F).*

•Figure 4: Lines 587-591 indicates a 2-fold increase in abundance that is not
apparent in the cell proportions indicated in Figure 4D. Also the numbers of each
subcluster indicated in the text in lines 589-593 do not match with the cell
proportions given in the table in Figure 4D.

We thank the reviewer for noticing this. The bar plot in figure 4D represents the three
main clusters (1-3) after cluster 0 was removed, as indicated in **line 840**. We have
now amended the text with the correct cell frequencies as follow:

**Line 856**: *These populations seemed dynamic over the course of infection, with*
*chronic stages associated with a 1.27- and 1.61-fold increase in the abundance of*
*Cd4⁺ T cells compared to other subclusters (23.64%, 30.21%, and 38.1% in naïve,*
*25dpi, and 45dpi, respectively) (Figure 4B and 4D), consistent with previous reports*
*^{26,27}. Of note, the subcluster identified as cluster 2 Cd8⁺ T cells (Cd8⁺ 2 T cells) was*
*only detected in infected samples but not in naïve controls (35.98% and 36.5% at 25*
*and 45dpi, respectively) (Figure 4B), indicating a disease-associated T cell subset in*
*the brain*

•Figure 5 A. Should Cluster 5 be Cluster 0?

Thank you for highlighting this. We have now amended cluster 5 and labelled it as
cluster 0 in the figure.

The percentages given in the text for the clusters in A do not match what is provided
in the figure panel 5A. For example, Cluster 1 = 34% (line 625) and Cluster 3 =
8.48% (lines 626). In the figure, there are fewer dots in cluster 1 than 3. There is no
description for Panel L.

In line with this comment, we have amended the labelling of the different clusters to
represent the corresponding percentages reported in the main text. We have also
added the following text to the figure caption describing the results presented in
panel L:

**Line 1433:** *L) qRT-PCR analysis of pro-inflammatory mediators ($Il1\beta$ and $Tnf\alpha$) in*
*BV2 microglia cell lines exposed to LPS in the presence of the B cell supernatant*
*with or without an anti-IL-10 blocking antibody. Pairwise comparisons were*
*conducted against cells exposed to LPS alone using Mann-Whitney test. A p values*
*<0.05 were considered statistically significant. * $p < 0.05$; ** $p < 0.005$; *** $p <$*
*0.0005.*

•Figure 6. How are the average and percent expression in panel D calculated?
This is calculated using the Seurat function “DotPlot” on the integrated layer, splitting
by experimental groups. The package automatically calculates the average
expression level and the % of cells expressing the gene(s) of interest.

•Figure S1. More information is needed to interpret the flow cytometry panels. Are
“infected” samples from 25 or 45 dpi? Macs and Micros are not defined. The live
dead assay is not provided. The gating strategy in C is not clear from the figure
legend. There is not enough information in the legend to interpret panel C. $Tnfsf13b$
and $Tnfrsf17$ are not shown.

Based on the reviewer’s comments, we have now amended this figure. We show
representative flow cytometry experiments of a typical single cell suspension from
murine hypothalamus from naïve (top) and infected samples (bottom) at 25dpi and
45dpi. We have amended the figure legend as follow:

**Line 1528: A)** *Representative flow cytometry analysis from naïve sample (top panel),*
*and infected samples at 25 (middle panel) and 45 days post-infection (bottom panel),*
*showing the relative proportion of macrophages ($Cd45^{High} CD11b^{high}$), microglia*
*($CD11b^{High} CD45^{low}$), oligodendrocytes ($O4^+$), and astrocytes ($ACSA2^+$) from the live*
*cells gate. B) Average number of genes (top) and transcripts (bottom) per cell in the*
*hypothalamic scRNAseq after filtering low quality cells, split by biological replicate. For*
*normalisation, we accounted for differential gene and UMI counts using two*
*independent approaches (SCT and STACAS). Both packages broadly identified the*
*same cell populations. C) Top 25 most highly variable genes identified by Scater. The*
*dotted line represents the median gene count. D) Gating strategy for the identification*
*of brain resident $CD45^+$ immune cells. Before processing, mice were inoculated i.v.*
*with anti- $CD45$ antibody conjugated with PE, labelling all circulating $CD45^+$ immune*
*cells. After lymphocyte preparation from brain samples using Percoll gradient, samples*
*were re-stained with anti- $CD45$ antibody conjugated with Brilliant Violet 421 labelling*
*tissue resident $CD45^+$ immune cells not previously exposed to anti- $CD45$ -PE. The*
*combination of these fluorophores allows to separate circulating from resident immune*
*cells. This gating strategy was used to identify brain-resident plasma cells, as well as*
*$BAFF^+$ microglia.*

•Figure S2. Could the transcript names be color coded to indicate which cluster they
are associated?

This is a great idea, and we thank the reviewer for this suggestion. We have
changed the colours of the marker genes based on their cluster of origin.

•Figure S3. It is unclear to me what the quality controls experiments are testing: the
integrity of the brain regions during processes or the reproducibility in the number of
transcriptional clusters detected? What should the reader expect to see if the data
pass quality control?

We apologise for any confusion on this matter. The standard QC metrics for the
analysis of 10X Visium datasets includes the analysis of gene density distribution per
spot, in addition to the sequencing depth, number of genes and number of
transcripts per spot. Some of these parameters are provided in Table S3A, as
referenced in the methods section. We have now amended figure S3 (and figure
legend) to include spatial feature plots depicting the overall gene density across the
tissue sections (figure S3A), in addition to the spatial transcriptional clusters
identified (Figure S3B), and the marker genes identifying anatomical regions in the
murine brain (Figure S3C). We have also changed the text in the methods section as
follow:

**Line 371:** *Downstream analyses of the expression matrices were conducted using*
*the Seurat pipeline for spatial RNA integration*^{28,29} *(Table S3A), and the overall gene*
*density per spot, as a quality control metric, in the different tissue sections is*
*reported in Figure S3A.*

Also, it is unclear to me what the 18 clusters represent. Are they the same clusters
as in Figures 1 and S5?

The 13-19 clusters shown in figure S3C represent distinct, spatially resolved
transcriptional clusters, that broadly coincide with specific anatomical locations in the
mouse brain (e.g, hippocampus, cortex, hypothalamus, thalamus). The clusters
identified in Figure 1 and S5 are different from the ones reported in Figure S3. We
have clarified this in the methods section as follow:

**Line 371:** *Downstream analyses of the expression matrices were conducted using*
*the Seurat pipeline for spatial RNA integration*^{28,29} **(Table S3A)**, *and the overall*
*gene density per spot, as a quality control metric, in the different tissue sections is*
*reported in **Figure S3A**. Specifically, the data was scaled using the SCTransform*
*function with default parameters. We then proceeded with dimensionality reduction*
*and clustering analysis using RunPCA (assay = "SCT"), FindNeighbours and*
*FindClusters functions with default settings and a total of 30 dimensions. We then*
*applied the FindSpatiallyVariables function to identify spatially variable genes, using*
*the top 1,000 most variable genes and "markvariogram" as selection method. The*
*approach enabled us to identify 13-19 distinct, spatially resolved transcriptional*
*clusters in the different tissue sections included in this study. We optimised the*
*parameters to obtain clustering of distinct spatially variable gene sets (**Figure S3B**)*
*that broadly coincide with several brain regions, including cortex, hippocampus, 3rd*
*and lateral ventricles, thalamus, hypothalamus, striatum, and amygdala (**Figure***
***S3C****), confirming the robustness, reproducibility, and reliability of our data.*

**References**

- 1. Brennan, F. H. *et al.* Microglia coordinate cellular interactions during spinal cord
repair in mice. *Nature Communications* **13**, 4096 (2022).
- 2. Stock, A. D. *et al.* Tertiary lymphoid structures in the choroid plexus in
neuropsychiatric lupus. *JCI Insight* **4**, (2019).
- 3. Moseman, E. A., Blanchard, A. C., Nayak, D. & McGavern, D. B. T cell engagement
of cross-presenting microglia protects the brain from a nasal virus infection. *Science*
*Immunology* **5**, (2020).
- 4. Subbarayan, M. S., Hudson, C., Moss, L. D., Nash, K. R. & Bickford, P. C. T cell
infiltration and upregulation of MHCII in microglia leads to accelerated neuronal loss
in an α -synuclein rat model of Parkinson's disease. *Journal of Neuroinflammation* **17**,
(2020).

- 5. Dong, Y. & Yong, V. W. When encephalitogenic T cells collaborate with microglia in
multiple sclerosis. *Nature Reviews Neurology* **15**, 704–717 (2019).
- 6. Jordão, M. J. C. *et al.* Neuroimmunology: Single-cell profiling identifies myeloid cell
subsets with distinct fates during neuroinflammation. *Science (1979)* **363**, (2019).
- 7. Van Hove, H. *et al.* A single-cell atlas of mouse brain macrophages reveals unique
transcriptional identities shaped by ontogeny and tissue environment. *Nature*
*Neuroscience* **22**, 1021–1035 (2019).
- 8. Surumbayeva, A. *et al.* Preparation of mouse pancreatic tumor for single-cell RNA
sequencing and analysis of the data. *STAR Protocols* **2**, (2021).
- 9. Romanov, R. A. *et al.* Molecular design of hypothalamus development. *Nature* (2020)
doi:10.1038/s41586-020-2266-0.
- 10. Chen, R., Wu, X., Jiang, L. & Zhang, Y. Single-Cell RNA-Seq Reveals Hypothalamic
Cell Diversity. *Cell Reports* **18**, 3227–3241 (2017).
- 11. Kim, D. W. *et al.* The cellular and molecular landscape of hypothalamic patterning and
differentiation from embryonic to late postnatal development. *Nature Communications*
**11**, (2020).
- 12. Yu, H., Rubinstein, M. & Low, M. J. Developmental single-cell transcriptomics of
hypothalamic POMC neurons reveal the genetic trajectories of multiple
neuropeptidergic phenotypes. *Elife* **11**, (2022).
- 13. Mickelsen, L. E. *et al.* Single-cell transcriptomic analysis of the lateral hypothalamic
area reveals molecularly distinct populations of inhibitory and excitatory neurons.
*Nature Neuroscience* **22**, 642–656 (2019).
- 14. Hajdarovic, K. H. *et al.* Single-cell analysis of the aging female mouse hypothalamus.
*Nature Aging* (2022) doi:10.1038/s43587-022-00246-4.
- 15. Olah, M. *et al.* Single cell RNA sequencing of human microglia uncovers a subset
associated with Alzheimer’s disease. *Nature Communications* **11**, (2020).
- 16. Pluvinae, J. v. *et al.* CD22 blockade restores homeostatic microglial phagocytosis in
ageing brains. *Nature* **568**, 187–192 (2019).
- 17. Nadjar, A., Wigren, H. K. M. & Tremblay, M. E. Roles of microglial phagocytosis and
inflammatory mediators in the pathophysiology of sleep disorders. *Frontiers in*
*Cellular Neuroscience* **11**, 1–11 (2017).
- 18. Bohlen, C. J., Friedman, B. A., Dejanovic, B. & Sheng, M. Microglia in Brain
Development, Homeostasis, and Neurodegeneration. *Annual Review of Genetics* **53**,
263–288 (2019).
- 19. Li, Q. & Barres, B. A. Microglia and macrophages in brain homeostasis and disease.
*Nature Reviews Immunology* vol. 18 225–242 Preprint at
<https://doi.org/10.1038/nri.2017.125> (2018).
- 20. Stuart, T. *et al.* Comprehensive Integration of Single-Cell Data. *Cell* **177**, 1888-
1902.e21 (2019).
- 21. Yip, S. H., Sham, P. C. & Wang, J. Evaluation of tools for highly variable gene
discovery from single-cell RNA-seq data. *Briefings in Bioinformatics* **20**, 1583–1589
(2018).
- 22. Townes, F. W., Hicks, S. C., Aryee, M. J. & Irizarry, R. A. Feature selection and
dimension reduction for single-cell RNA-Seq based on a multinomial model. *Genome*
*Biology* **20**, (2019).
- 23. Yip, S. H., Sham, P. C. & Wang, J. Evaluation of tools for highly variable gene
discovery from single-cell RNA-seq data. *Briefings in Bioinformatics* **20**, 1583–1589
(2018).

- 24. McCarthy, D. J., Campbell, K. R., Lun, A. T. L. & Wills, Q. F. Scater: Pre-processing,
quality control, normalization and visualization of single-cell RNA-seq data in R.
*Bioinformatics* **33**, 1179–1186 (2017).
- 25. Fujita, A. *et al.* Hypothalamic tuberomammillary nucleus neurons:
Electrophysiological diversity and essential role in arousal stability. *Journal of*
*Neuroscience* **37**, 9574–9592 (2017).
- 26. Laperchia, C. *et al.* Trypanosoma brucei Invasion and T-cell Infiltration of the Brain
Parenchyma in Experimental Sleeping Sickness: Timing and Correlation with
Functional Changes. *PLoS Neglected Tropical Diseases* **10**, 1–19 (2016).
- 27. Lyck, R. *et al.* T-cell interaction with ICAM-1/ICAM-2 double-deficient brain
endothelium in vitro: The cytoplasmic tail of endothelial ICAM-1 is necessary for
transendothelial migration of T cells. *Blood* **102**, 3675–3683 (2003).
- 28. Stuart, T. *et al.* Comprehensive Integration of Single-Cell Data. *Cell* **177**, 1888-
1902.e21 (2019).
- 29. Hao, Y. *et al.* Integrated analysis of multimodal single-cell data. *Cell* **184**, 3573-
3587.e29 (2021).

REVIEWERS' COMMENTS

Reviewer #1 (Remarks to the Author):

In this revised manuscript, the authors address many of the concerns raised in the initial review. Two primary concerns remain.

1) In response to a concern raised about the ontogeny of the "infection-associated microglia" cluster, the authors state in their point-by-point response: we cannot decipher the ontogeny of these cells with the current dataset with enough certainty. We have renamed the Aif1+ clusters as "Infection-associated microglia" (IAM) and the Cx3cr1+ clusters as "homeostatic microglia" throughout the text to better reflect the underlying complexity of these myeloid subsets.

For the reasons stated by the authors, if they can't discern the ontogeny of the cell types in the clusters, they should consider renaming the clusters as myeloid cells to be accurate. It is well known that microglia and macrophages express Iba-1 and CX3CR1 during inflammation. It is incredibly important to be accurate on these points.

2) In response to a question regarding the anti-inflammatory nature of the mrc1+ cluster, the authors provide additional context that they did not find canonical anti-inflammatory genes in the cluster and also found pro-inflammatory gene expression (ilb), yet continue to describe the cluster as anti-inflammatory in the text and do not include the additional information on ilb and il18bp expression.

Overall, the characterization of the clusters needs to be improved to not be misleading or inaccurate. This should require modifications to the text only.

Reviewer #2 (Remarks to the Author):

The authors have addressed all concerns.

Reviewer #3 (Remarks to the Author):

I appreciate the time and work the authors put into responding to my review. They have adequately addressed my concerns.

**Decision on Nature Communications manuscript NCOMMS-22-13214**

Once again, we thank all the reviewers for their positive assessments and helpful
comments.

Reviewer #1 (Remarks to the Author):

In this revised manuscript, the authors address many of the concerns raised in the initial
review. Two primary concerns remain.

1) In response to a concern raised about the ontogeny of the “infection-associated
microglia” cluster, the authors state in their point-by-point response: we cannot decipher
the ontogeny of these cells with the current dataset with enough certainty. We have
renamed the Aif1+ clusters as “Infection-associated microglia” (IAM) and the Cx3cr1+
clusters as “homeostatic microglia” throughout the text to better reflect the underlying
complexity of these myeloid subsets.

For the reasons stated by the authors, if they can't discern the ontogeny of the cell types
in the clusters, they should consider renaming the clusters as myeloid cells to be
accurate. It is well known that microglia and macrophages express Iba-1 and CX3CR1
during inflammation. It is incredibly important to be accurate on these points.

We agree that the only way to directly assess the ontogeny of these various cell types is
by using lineage tracing approaches. Based on the transcriptional signatures of these
clusters we were able to identify several “infection-associated” subsets that are identical
to the ones previously reported for “disease-associated microglia” in neurodegenerative

disorders, including high expression levels of *Apoe*, *Itgax* (encoding CD11c), and *Cst7*, and low or undetectable expression of signatures associated with homeostatic microglia such as *Tmem119*, *P2ry12*, and *Cx3cr1*^{1,2}, just to name a few. When examining the infection-associated subclusters in more detail, we noted that the cluster “Infection-associated 1”, and to a lesser extent cluster “Infection-associated 2” express low levels of *Tmem119*, *P2ry12*, and *Cx3cr1*, and higher levels of *Apoe*, *Itgax*, and *Cst7* (**Figure 1**), strongly suggesting their microglia origin. However, to ensure we adequately reflect these limitations, we have renamed these clusters are “infection-associated mononuclear phagocytes” throughout the text.

We have amended the text in the manuscript to clarify that although we speculate that these are microglia cells, further studies are required to clarify their ontogeny.

*Line 640: Based on these results we catalogued*
*clusters 0 to 5 as follow: Homeostatic microglia (HM) 1 (1,688 cells; 26.77%), HM 2,*
*(1,548 cells; 24.55%), Cd14+ Monocytes (1,396 cells; 22.14%),Mrc1+ border-*
*associated macrophages (Mrc1+ BAMs – 812 cells; 12.6%), Infection-associated*
*mononuclear phagocytes (IAMNP) 1 (587 cells; 9.31%), and IAMNP 2 (274 cells;*
*4.34%) (Figure 3C and D).*

*Line 1081: Although we propose that the two “infection-associated” myeloid subsets*
*identified are likely to be microglia based on their similarities to previously reported*

*disease-associated microglia, we cannot exclude the possibility that they might also*
*be derived from perivascular or peripheral myeloid cells that engraft in the brain in*
*response to chronic inflammation*

2) In response to a question regarding the anti-inflammatory nature of the *mrc1+* cluster,
the authors provide additional context that they did not find canonical anti-inflammatory
genes in the cluster and also found pro-inflammatory gene expression (*ilb*), yet continue
to describe the cluster as anti-inflammatory in the text and do not include the additional
information on *ilb* and *il18bp* expression.

Overall, the characterization of the clusters needs to be improved to not be misleading or
inaccurate. This should require modifications to the text only.

We thank this reviewer for raising this point. The *Mrc1+* cluster does express *Il1rn* which
inhibits the activity of IL-1 α and IL-1 β as well as *Il18bp* which can act as a decoy
receptor for IL-18 (see table S2F), which together with *Chil3*, *Arg1*, and *Vegfa* are
putatively associated with anti-inflammatory/tissue repair macrophages. We have
included these two additional genes in figures 3A and 3B, and have also amended the
text in the manuscript as follow:

Line 623: *Cluster 3 expresses putative marker genes associated with border-*
*associated macrophages such as *Lyz2*, *Ms4a7*, *Ms4a6c*, *Tgfb1*, *H2-Ab1*, and *Lyz2**
*^{6,7}, as well as gene sets characteristic of anti-inflammatory responses, such as *Mrc1**
*(encoding for CD206), *Chil3*, *Arg1*, *Il1rn*, *Il18bp*, and *Vegfa* (Figure 3B and S2F*
*Table), indicative of an anti-inflammatory phenotype*

Reviewer #2 (Remarks to the Author):

The authors have addressed all concerns.

We thank this reviewer once again for helpful revisions and discussions. It has greatly
improved our manuscript.

Reviewer #3 (Remarks to the Author):

I appreciate the time and work the authors put into responding to my review. They have
adequately addressed my concerns.

We thank this reviewer for their positive appreciation of our work and for raising
important questions around our work. We believe the final manuscript is much improved
after peer-review.

References

- 1. Olah, M. *et al.* Single cell RNA sequencing of human microglia uncovers a
subset associated with Alzheimer's disease. *Nat Commun* **11**, (2020).
- 2. Deczkowska, A. *et al.* Disease-Associated Microglia: A Universal Immune
Sensor of Neurodegeneration. *Cell* vol. 173 1073–1081 Preprint at
<https://doi.org/10.1016/j.cell.2018.05.003> (2018).